# Does Entity Abstraction Help Generative Transformers Reason?

**Nicolas Gontier**                                                        *gontiern@mila.quebec*
*Quebec Artificial Intelligence Institute (Mila), Montreal, Canada*
*Polytechnique Montreal, Canada*
*ServiceNow Research*

**Siva Reddy**                                                              *siva.reddy@mila.quebec*
*Quebec Artificial Intelligence Institute (Mila), Montreal, Canada*
*McGill University, Montreal, Canada*
*Facebook CIFAR AI Chair*
*ServiceNow Research*

**Christopher Pal**                                                        *christopher.pal@mila.quebec*
*Quebec Artificial Intelligence Institute (Mila), Montreal, Canada*
*Polytechnique Montreal, Canada*
*Canada CIFAR AI Chair*
*ServiceNow Research*

**Reviewed on OpenReview:** *https://openreview.net/forum?id=9nhmKwLAWV*

## Abstract

We study the utility of incorporating entity type abstractions into pre-trained Transformers and test these methods on four NLP tasks requiring different forms of logical reasoning: (1) compositional language understanding with text-based relational reasoning (CLUTRR), (2) abductive reasoning (ProofWriter), (3) multi-hop question answering (HotpotQA), and (4) conversational question answering (CoQA). We propose and empirically explore three ways to add such abstraction: (i) as additional input embeddings, (ii) as a separate sequence to encode, and (iii) as an auxiliary prediction task for the model. Overall, our analysis demonstrates that models with abstract entity knowledge performs better than without it. The best abstraction aware models achieved an overall accuracy of 88.8% and 91.8% compared to the baseline model achieving 62.9% and 89.8% on CLUTRR and ProofWriter respectively. However, for HotpotQA and CoQA, we find that F1 scores improve by only 0.5% on average. Our results suggest that the benefit of explicit abstraction is significant in formally defined logical reasoning settings requiring many reasoning hops, but point to the notion that it is less beneficial for NLP tasks having less formal logical structure.

## 1 Introduction

Transformer language models (TLMs; Vaswani et al. 2017) have enabled rapid progress in natural language processing (NLP). When pre-trained on large corpora (such as the web) to predict the next tokens or a set of masked tokens from an input sequence, TLMs can capture linguistic knowledge (Peters et al., 2018; Goldberg, 2019; Tenney et al., 2019b), and yield state-of-the-art performance on many NLP tasks with little to no task supervision (Devlin et al., 2019; Radford et al., 2018; 2019; Brown et al., 2020). However, it is not clear if these models can capture higher level knowledge such as reasoning skills that can be re-used in arbitrary contexts, and in ways that leverage the compositionality of those skills (Lake & Baroni, 2018; Liška et al., 2018), something logical reasoners can do relatively well on a smaller scale (De Raedt et al., 2007; Fierens et al., 2015). Simple compositional tasks such as SCAN (Lake & Baroni, 2018), CLUTRR (Sinha

et al., 2019), and ProofWritter (Clark et al., 2020; Tafjord et al., 2021) can help diagnose the compositional generalization behavior of language models. Recent work on some of these datasets showed that TLMs still struggle to learn reasoning strategies that can be re-used in out-of training distribution settings (Lake & Baroni, 2018; Gontier et al., 2020).

Humans do abstraction to simplify reasoning and become very efficient. This is particularly true in mathematics when manipulating variables instead of numbers. Manipulating abstract concepts allows humans to generalize knowledge across domains. If we look at how logical reasoners operate, we find that they have an important abstraction component (going from grounded entities to higher level concepts) before logical reasoning can start (De Raedt et al., 2007). Going from an original text sequence to its higher-order meaning is an important part of the NLP pipeline (part of it being entity type tagging). Similarly in mathematics, the introduction of generic variables allows to progress in a logical reasoning process without keeping track of every (grounded) atomic entity. Overall, this idea that we call abstraction, seems to be an important part of logical reasoning. Recent work suggests that incorporating external knowledge about grounded entities could improve language models' abilities to reason and generalize (Ji et al., 2017; Zhang et al., 2019; Moosavi et al., 2020; Rosset et al., 2020). However the empirical effect of incorporating generic entity types remains unclear, especially with recent studies suggesting that pre-trained models already encode some of that linguistic knowledge in their parameters (Hewitt & Manning, 2019; Liu et al., 2019a; Clark et al., 2019; Tenney et al., 2019a). In this work, we study the effect of explicitly providing entity type abstraction *in addition to* the original input to pre-trained Transformers.

We explore and evaluate different ways to incorporate entity type abstraction and observe that some methods are more effective than others. To construct the abstract representation incorporated into TLMs, we leverage entity type information given by fixed pre-trained models. This allows for automatic and reproducible data processing. In general, our approach is the following: given an input sequence, we use an entity tagger to label entity types in the sequence. We then use these labels to construct a copy of the original sequence in which all entities are replaced by their corresponding entity types. This new sequence can then be used as extra input (Sections 3.1 & 3.2) or as extra training signal to the model (Sections 3.3). In particular, we explore three different ways to augment pre-trained Transformers with this abstract knowledge:

1. by combining token embeddings from both the original and the abstract sequence before encoding (Section 3.1) (Figure 1a & 1b).
2. by encoding both the original and the abstract sequence and combining them before decoding the target output (Section 3.2) (Figure 1c & 1d).
3. by adding a second language model head on top of the Transformer decoder to predict the abstract sequence (Section 3.3) (Figure 1e).

A series of controlled experiments on two synthetic datasets show that models having access to abstract knowledge about entity types yield better performance at inference time both when interpolating and extrapolating to unseen lengths of reasoning chains. Since no (non-synthetic) natural language dataset systematically and explicitly requires long chains of reasoning, we used synthetic datasets in which we can control for the degree of compositional generalization required. Nevertheless, in order to understand if the benefits observed could also be applicable in more realistic settings, we ran a series of experiments on two question answering datasets requiring some degrees of multi-hop reasoning. Results on these more natural language datasets show that abstraction aware models are not significantly better than baseline models. We conclude that these "real-world" problems do not have strong enough logical structure to benefit from the abstraction technique and that the pre-trained weights of large language models seem to be "enough" for such natural language tasks, confirming previous results (Goldberg, 2019; Tenney et al., 2019a). It is only when tasked on logical problems that explicitly require reasoning depths unseen during training that abstraction becomes significantly beneficial.

Overall our work contributions are the following:

1. we introduce and compare empirically different ways to incorporate abstraction into pre-trained TLMs.
2. we show that incorporating abstract knowledge can significantly improve compositional generalization to unseen lengths of reasoning chains in multi-step reasoning tasks.

3. we show that abstraction aware models may not benefit much when language is more natural and less procedural.

We hope that our work will inspire future research in the field to look for simple inductive biases that can complement pre-trained models in their quest to achieve logical reasoning at scale.

## 2 Related Work

Augmenting neural language models with knowledge about entities has been a popular method to improve their functionality. Ji et al. (2017) trained an entity neural language model to predict sequences of entities with an LSTM (Hochreiter & Schmidhuber, 1997). At each sampling step, they predict the next word alongside a categorical variable indicating the current token's entity ID. They obtained lower perplexity and better results on co-reference resolution and entity prediction tasks than a variety of baselines. Similarly, Rosset et al. (2020) trained a GPT2 model (Radford et al., 2019) by giving it access to entity knowledge at the input level and as an additional pre-training loss. Their model achieved better factual correctness on benchmarks such as LAMA (Petroni et al., 2019), and performed better than a baseline GPT2 model in various question answering tasks. Inspired by this work and motivated by the goal of building better reasoning language models, we instead focus on the prediction of entity *types* rather than entity *identifiers* taken from a fixed list of entities. This allows our solution to be robust to new entities. In addition, we explore and compare different ways to incorporate the entity knowledge in an encoder-decoder architecture.

Besides entity knowledge, other types of explicit information has also been given to language models. Prior work by Swayamdipta et al. (2018); Eriguchi et al. (2017); Nădejde et al. (2017) tried incorporating syntax information into language models by introducing an auxiliary loss to the model. Results show that models trained to also predict syntactic information achieved stronger performances on various tasks such as PropBank semantics and Neural Machine Translation. Inspired by this work, we also introduce an auxiliary loss but to predict entity types, and with an application towards reasoning tasks. Also working with syntax information, Sundararaman et al. (2019) incorporated POS tags into the input embedding of a BERT model. Results show improved BLEU score on machine translation and higher accuracy than baselines on the GLUE benchmark (Wang et al., 2018). Similarly, Sachan et al. (2021) augmented a pre-trained BERT model with a syntax graph neural network in order to encode syntax trees. Their results show that the quality of the trees are highly tied to the performance boost observed. Levine et al. (2020) trained a BERT-like model to learn word senses. They gave their model access to WordNet supersenses at the input level and as an additional training loss. They achieve better performance than other baselines on the SemEval Word Sense Disambiguation task (Raganato et al., 2017). Moosavi et al. (2020) propose to improve robustness to data biases by augmenting the training data with predicate-argument structures. They train a BERT-base model (Devlin et al., 2019) with PropBank-style semantic role labeling (Shi & Lin, 2019) on MultiNLI (Williams et al., 2018) and SWAG (Zellers et al., 2018) datasets. Their results show that incorporating predicate-argument structure in the input sequence (only during training) makes the model more robust to adversarial examples in MultiNLI. More recently, Porada et al. (2021) extended a RoBERTa model (Liu et al., 2019b) with hypernym abstraction based on WordNet to evaluate the plausibility of events. Their model is able to better predict human plausibility judgement than other RoBERTa baselines. **Although different in application, all these prior works leverage the general idea of explicitly giving more abstract knowledge to language models, hence showing how flexible and generic this strategy can be.** We take a similar approach with entity types, but in the hope of improving the reasoning skills of our baseline model.

A flurry of recent work has also examined ways to augment TLMs with entities from external knowledge bases (Zhang et al., 2019; Peters et al., 2019; Févry et al., 2020; Verga et al., 2020). However, most of the time, these solutions rely on external components such as knowledge graphs with pre-trained entity embeddings, and/or an additional memory. While they often use entity linking as a way to perform co-reference resolution, they do not incorporate higher level of abstractions such as entity types like we do here.

Table 1: Different architectures for different abstraction strategies. $X$ (blue) is the original sequence embedding, $X_s$ (green) is the embedding of the simplified sequence with entities replaced by their entity type tags, "*ENC*" is the T5 encoder, $H$ (blue) is the contextualized representation of sequence $X$, $H_s$ (green) is the contextualized representation of sequence $X_s$, "*DEC*" is the T5 decoder, and $Y$ is the target sequence to predict.

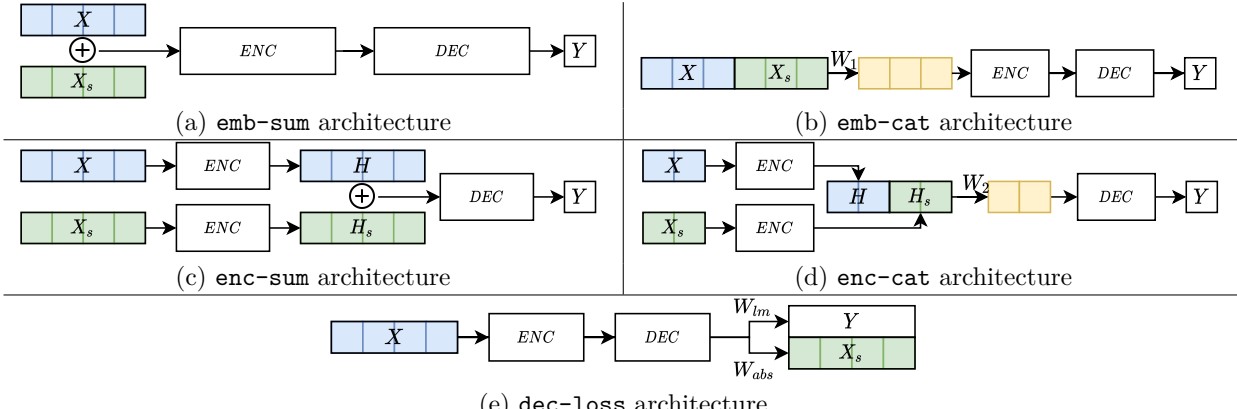

(a) `emb-sum` architecture

(b) `emb-cat` architecture

(c) `enc-sum` architecture

(d) `enc-cat` architecture

(e) `dec-loss` architecture

## 3 Introducing Abstraction Inductive Biases

In this section we describe five different ways to incorporate abstraction into a pre-trained encoder-decoder model. Given an input sequence $X$, we use `spacy` named entity tagger[1] to make a simplified copy $X_s$ of the input. This is a more generic copy of $X$.

We run the `spacy` recognizer on $X$ to extract entity tags such as *PERSON*, *ORG*, *GPE*, etc... For each entity type, we create $n$ additional vocabulary entries (with randomly initialized embeddings) such as [`PERSON_1`, ..., `PERSON_n`, `ORG_1`, ..., `ORG_n`, ...]. Every token in $X$ is then replaced by their (randomly numbered) entity tag to make the simplified sequence $X_s$. If the same entity is present multiple times in $X$, each occurrence will be replaced by the same entity tag in $X_s$. If no entity is found for a token in $X$, the original token's text is kept in $X_s$ (*e.g.* "*Bob Smith has a cat that he loves. Bob also loves Alexandra.*" would be transformed into "*PERSON_11 has a cat that he loves. PERSON_11 also loves PERSON_3.*").

We select $n$ greater than one as we believe that for some tasks it is important to differentiate between two entities of the same type. At the same time we select $n$ to be much smaller than the total number of entities in the dataset, thus forcing abstraction. In particular, the hyper-parameter $n$ is set to the smallest possible value such that each distinct entity within the *same* example gets a different entity tag. This is different for each dataset depending on the number of unique entities per example. Individual values can be seen in Appendix A. If the same entity appears more than once in a single example, it will get the same tag every time within that example. Since we re-use the same finite set of entity tags across all examples, each entity tag will be used for different entity tokens, thus after seeing many examples, entity tags of the same type will likely have a similar embedding. We discuss some result highlighting this phenomenon in Appendix B.

In order to better understand the influence of having multiple tags for the same entity type, we also ran experiments in which we set $n = 1$, thus forcing all entities of the same type to be mapped to the same embedding. However we noticed both more variance and weaker performance of our models, so the rest of this work will focus on the $n > 1$ setting described above. Results can be seen in Appendix C.

In the following subsections, we describe different strategies to incorporate $X_s$ into an encoder-decoder Transformer model.

### 3.1 Abstraction as an additional embedding

Our first strategy is to combine $X_s$ with $X$ at the embedding level. To do that, we construct $X_s$ to be of the same length of $X$. If a `spacy` entity spans over multiple tokens (*e.g.* ["*Alex*", "*andra*", "*Smith*" "*is*", "*the*", "*wife*", "*of*", "*Bob*"]), we copy its entity tag at each sub-token positions (*e.g.* ["*PERSON_3*", "*PERSON_3*",

---

[1] https://spacy.io/models/en

"*PERSON_3*", "*is*", "*the*", "*wife*", "*of*", "*PERSON_11*"]). For each token within both sequences we either sum (`emb-sum` experiments) or concatenate (`emb-cat` experiments) their respective token embeddings.

**sum**. In `emb-sum` experiments (Figure 1a), if tokens do not have entity tags associated with them, we ignore their embedding to avoid summing the same embedding twice for non-entity tokens. We only sum embeddings of tokens that do have an abstract tag associated with them. This is to ensure that we only modify pre-trained embeddings that correspond to entity tokens. This is done by masking out the tokens in $X_s$ that are the same in $X$. The input given to the model's encoder is then $emb(X) + mask \times emb(X_s) + positional$ with $emb()$ being the embedding matrix, $mask$ defined as the $X \neq X_s$ binary tensor, and $positional$ being the regular Transformer positional embedding. This resembles the setting used by Rosset et al. (2020), however their knowledge-aware embedding comes from a sequence of entities from a dictionary lookup, rather than a sequence of entity *types* from a pre-trained tagger. The advantage of our method is that it is robust to unseen entities of the same type.

**concat**. In `emb-cat` experiments (Figure 1b), if tokens do not have an entity tag associated with them, we replace their embedding with a generic (learnable) "$<grounded>$" token embedding. This ensures that all non-entity token embeddings gets modified in the same way compared to entity tokens. We eventually resize the concatenated embeddings with a learnable matrix $W_1 \in R^{2e \times e}$ with $e$ being the model's embedding size. The input given to the model's encoder is then $[emb(X); emb(X_s)] \cdot W_1 + positional$ with $emb()$ being the embedding matrix, $[;]$ defined as the concatenation operator, and $positional$ being the regular Transformer positional embedding. In this setup the model has $2e^2 + e$ more parameters.

### 3.2 Abstraction as an additional sequence to encode

Our second strategy is to combine $X_s$ with $X$ at the encoding level. To do that, we again construct $X_s$ to be of the same length of $X$. Similarly as above, if a `spacy` entity spans over multiple tokens, we copy its entity tag at each sub-token positions. We then encode both $X$ and $X_s$ with the same encoder weights to have two contextualized encodings: $H$ and $H_s$ respectively. For each token within both sequences we either sum (`enc-sum` experiments) or concatenate (`enc-cat` experiments) their respective contextualized encodings.

**sum**. For `enc-sum` experiments (Figure 1c), the input given to the model's decoder becomes $H + H_s$. Unlike in Section 3.1, in these experiments we do not mask any position because the encodings are contextualized over the entire sequence. All token representations were influenced by all other tokens because of the Transformer encoder attention mask. Thus even non-entity token representations were influenced by entity tokens.

**concat**. For `enc-cat` experiments (Figure 1d), we introduce a learnable matrix $W_2 \in R^{2d \times d}$ with $d$ being the model's encoding size in order to resize the concatenated encodings, similarly to Section 3.1. The input given to the model's decoder is then $[H; H_s] \cdot W_2$ with $[;]$ defined as the concatenation operator. In this setup the model has $2d^2$ more parameters.

### 3.3 Abstraction as an auxiliary task

Our third strategy is to incorporate $X_s$ into the model with an additional cross entropy loss (`dec-loss` experiments). The model will now be tasked to predict both the target output $Y$ as well as the abstracted input $X_s$. To do that, we introduce a second language model head $W_{abs} \in R^{d \times vocab}$ (with $d$ being the model's decoder size) on top of the model's decoder, initialized to have the same weights than the original language model head $W_{lm}$, and fine-tuned during training with a second cross-entropy loss. The final model's loss is then the average between the two cross entropy losses:

$$0.5 * \frac{1}{|X_s|} \sum_{i=0}^{|X_s|} P(X_s)_i * log(\text{softmax}(H^{dec} \cdot W_{abs}))_i$$

$$+0.5 * \frac{1}{|Y|} \sum_{j=0}^{|Y|} P(Y)_j * log(\text{softmax}(H^{dec} \cdot W_{lm}))_j,$$

with $H^{dec}$ being the output tensor of the model's decoder. Note that $X_s$ and $Y$ have different lengths ($|X_s|$ and $|Y|$ respectively), which is why we differentiate indices between the two sums. By sharing the decoder weights (except for the additional language model head), we make sure that most of the network parameters are influenced by the additional cross-entropy loss. This acts as a regularizer and forces the model to "*know*" about entity types within its original parameters. In this setup, the model has $d * vocab$ more parameters. Figure 1e illustrates this strategy. While not in the scope of our initial experiments, we evaluated the performance of this model to predict abstract sequences with its additional language model head. Curious readers can refer to Appendix B for more details.

## 4 Experiments

In this section we describe the experiments we ran on various datasets. We start with the CLUTRR benchmark (Sinha et al., 2019) in Section 4.1 as controlled experiments in which we know how much compositional generalisation is required. We then test our models on the ProofWriter (Clark et al., 2020; Tafjord et al., 2021) dataset in Section 4.2. This allows to verify if entity type abstraction is beneficial in formally defined logical reasoning environments with simple language. We next report experiments on the multi-hop question answering task HotpotQA (Yang et al., 2018) in Section 4.3. This allows to test if entity type abstraction is beneficial in two-hop question answering settings with more natural language which is by nature more noisy. Eventually, we also report results on the conversational question answering task CoQA (Reddy et al., 2019) in Section 4.4. This allows to test if entity type abstraction is beneficial in conversational settings in which the entity being discussed may be originally introduced much earlier in the conversation, thus requiring some entity linking before answering.

For all experiments we trained one model for each of the 5 different strategies presented in Section 3, in addition to a baseline model fine-tuned without any abstraction knowledge. We used the `AllenNLP` library (Gardner et al., 2017) with the HuggingFace `transformers` library (Wolf et al., 2019) PyTorch implementation of `T5-small` with 16-bit floating point precision. Each experiment was run on tesla V100 32gb GPUs with early stopping and a patience of 10 epochs on the validation set (defined as a 10% split of the training set). We report all hyper-parameters and library versions in Appendix A for reproducibility purposes.

### 4.1 Compositional generalization with CLUTRR

Although synthetic, CLUTRR is used because it allows for controlled experiments in which we can clearly measure both interpolation and extrapolation performance of our model. We generated $390,000$ examples that were roughly split 77/23 between training and testing. Each example consists of a unique (non-cyclic) family graph. The goal of this task is to infer the type of edge (family relation) between two nodes (two entities) that are the further apart in the input graph. The bigger the graph, the more hops are required to infer the missing edge. We express each family graph, along with its question and answer in text using a simple "*[e_1] is the [rel] of [e_2]*" template. Some examples of input/output sequence pairs can be seen in Appendix D.

We evaluate the generated answer accuracy. The answer is defined as the first sentence in the generated sequence. Since all answers during training were expressed using a simple template, we inverse this template to extract the $(\hat{e}1, \hat{rel}, \hat{e}2)$ triple from the generated answer. If the extraction fails, we consider the generated answer wrong. We then compare the extracted triple to the ground truth provided by the CLUTRR dataset. If the reference solution is $(e1, rel, e2)$, we accept both $(e1, rel, e2)$ and $(e2, inv\_rel, e1)$ as valid solutions, with $inv\_rel$ being the inverse relation of $rel$. For instance, the inverse relation of "*father*" can be "*son*" or "*daughter*", we accept both.

### 4.1.1 Testing Compositional generalization

We carefully divided train and test sets to force the model to generalize to both unseen graph sizes (*i.e.*: unseen reasoning depth) and unseen $(e1, rel, e2)$ triples. Specifically, the training set is made of graphs with 2, 4 and 6 relations between 3, 5 and 7 entities respectively, while the test set is made of graphs with up to

Table 2: Prediction accuracy on CLUTRR test set for all difficulty levels. Models have been trained on levels 2, 4, 6 with only 9.58% of all the $(e1, rel, e2)$ triples present in the test set. Per-level performance is colored in shades of green for better visualisation. The red boxed area indicates test problems at depths unseen during training.

| CLUTRR | 2 | 3 | 4 | 5 | 6 | 7 | 8 | 9 | 10 | avg. |
|---|---|---|---|---|---|---|---|---|---|---|
| no abstraction | 100% | 84.4% | 64.8% | 61.5% | 54.2% | 56.6% | 45.5% | 42.6% | 56.8% | **62.9%**(±0.18) |
| emb-sum | 100% | 85.3% | 74.6% | 59.3% | 64.3% | 67.7% | 65.1% | 58.9% | 68.4% | **71.5%**(±0.13) |
| emb-cat | 94.6% | 60.0% | 35.6% | 42.0% | 40.2% | 74.3% | 78.2% | 77.9% | 80.6 | **64.8%**(±0.21) |
| enc-sum | **100%** | **94.8%** | **86.9%** | **89.9%** | **85.6%** | **87.2%** | **85.1%** | **84.3%** | **85.5%** | **88.8%**(±0.05) |
| enc-cat | 100% | 86.1% | 72.6% | 70.2% | 66.7% | 69.1% | 63.8% | 61.9% | 74.0% | **73.8%**(±0.12) |
| dec-loss | 100% | 74.7% | 64.9% | 59.3% | 56.2% | 61.3% | 52.8% | 47.8% | 61.3% | **64.2%**(±0.15) |

10 relations between 11 entities. In addition, all possible entities and relations are seen during training but only 9.58% of $(e1, rel, e2)$ triples from the test set are also in the training set. This small overlap makes the test set harder than originally designed and allows to analyse the compositional generalization capacity of our model.

We fine-tuned a `T5-small` model on $300,000$ training examples of levels $2, 4$ and $6$ and evaluate the model on 9 test sets of $10,000$ examples each for all levels from 2 to 10. The level is defined as the number of edges (relations) in the graph. The higher the level is, the bigger the graph is, the further apart the two entities to link are, and the greater the number of hops required to answer the query is. Specifically, we test levels $2, 4$ and $6$ for compositional generalization, levels 3 and 5 for interpolation, and levels $7, 8, 9$ and 10 for extrapolation.

### 4.1.2 Results

Table 2 shows answer accuracy on each test set level for models trained with $n = 20$ tokens per entity type. As mentioned in Section 3, this helps keeping track of "who's who" in the abstracted sequence, which is important for solving a task such as CLUTRR. We see that the best model (`enc-sum`) strongly outperforms all other models with an average score of 88.8% compared to 62.9% for the baseline.

**sum -vs- concatenation**. We can see that for both the embedding strategies (`emb-cat` & `emb-sum`) and the encoding strategies (`enc-cat` & `enc-sum`), summing representations together yields better performance than feeding their concatenation through a linear layer. One hypothesis is that in the `emb-cat` model, all pre-trained embeddings are modified by either a entity type token embedding or the "<grounded>" embedding, whereas the `emb-sum` model keeps most of its pre-trained embeddings unchanged. This result also suggests that sometimes, simpler methods are more effective.

**embedding -vs- encoding**. From Table 2 we also see that, on average, processing the abstract sequence through the encoder yields better performance than only processing it through the token embedder. This is expected as more layers of the Transformer can process the abstracted sequence.

**learning the abstraction**. In the last line of Table 2 the abstraction is given as output to the model. This experiment tests if learning how to abstract along-side the original task helps the model. We can see that learning to predict the abstract sequence can indeed help generalize as the average score increases from 62.9% for the baseline to 64.2% for the `dec-loss` model. However it is not the most effective method in this case. We believe this is because we test longer reasoning lengths (up to 10 steps), making the input sequence much longer than what is seen during training (up to 6 steps). This is because in CLUTRR, to make long reasoning chains the input family story gets longer. This penalizes the `dec-loss` model since it is designed to generate the abstracted version of the input, regardless of its length.

Overall, we can see that abstraction-aware models can all extrapolate better than the baseline model, suggesting that entity abstraction does help pre-trained Transformers to compositionally generalize. To verify that this is not just a feature of the CLUTRR dataset, we perform the same analysis on another synthetic dataset in the next section.

Table 3: Prediction accuracy on different slices of the ProofWriter D5 test set for all our models and the originally reported numbers by Clark et al. (2020). Models have been trained on depth D0, D1, D2. Models are trained in the "open-world" assumption (OWA), except for the original Clark et al. (2020) model which was trained in the "closed-world" assumption (CWA). Per-depth performance is colored in shades of green for better visualisation. The red boxed area indicates test problems at depths unseen during training.

| ProofWriter | RoBERTa-large Clark et al. (2020) CWA | no abstraction OWA | emb-sum OWA | emb-cat OWA | enc-sum OWA | enc-cat OWA | dec-loss OWA |
|---|---|---|---|---|---|---|---|
| Overall | 83.9%(±0.29) | 89.8%(±0.11) | 90.9%(±0.07) | 90.9%(±0.09) | 88.4%(±0.08) | 90.1%(±0.07) | **91.8%**(±0.07) |
| D0 | **100.0%** | 99.5% | 99.2% | 99.5% | 98.9% | 99.2% | 99.4% |
| D1 | **98.8%** | 95.6% | 93.5% | 95.3% | 89.5% | 91.8% | 95.1% |
| D2 | **98.8%** | 87.9% | 81.0% | 87.1% | 75.3% | 83.5% | 86.6% |
| D3 | 71.1% | 83.7% | 84.6% | 85.9% | 81.7% | 85.2% | **87.4%** |
| D4 | 43.4% | 77.3% | **87.4%** | 82.2% | 84.7% | 84.1% | 85.0% |
| D5 | 37.2% | 70.0% | **85.3%** | 74.8% | 84.0% | 79.6% | 80.6% |

## 4.2 Abductive Reasoning with ProofWriter

Similarly to CLUTRR, the ProofWriter dataset (Tafjord et al., 2021) is a collection of synthetic facts and rules with derived conclusions. The goal of this task is to infer the truth value of an unknown statement given a series of known facts and 1-hop inference rules. Each example is made of a different set of facts, rules, and an unknown statement. The model then has to predict if the unknown statement is "*True*", "*False*", or "*Unknown*" according to the input knowledge. Each fact and rule is expressed in simple templated language given by a grammar. Some examples of input/output pairs can be seen in Appendix D.

The dataset also provides the required chain of inference required to arrive at the final answer. The number of such 1-hop inference steps is considered the "depth" of the example. For instance a depth-0 (D0) example simply requires to see if the unknown statement is present in the input list of fact or not; a depth-1 (D1) example requires to apply one inference rule to one fact to arrive at the answer; etc... The dataset contains examples of up to depth-5 (D5) inference chains. We test for compositional generalisation by training on examples of up to depth 2 reasoning chains (D0, D1, D2) and testing on examples for each depth from D0 to D5.

We fine-tuned `T5-small` models on the official training and development set from the depth $<= 2$ data folder and tested it on the test set from the depth $<= 5$ data folder; consisting of 70,076 training examples and 20,030 testing examples. We trained one model for each of the 5 different strategies presented in Section 3, in addition to a baseline model fine-tuned without any abstraction knowledge. Unlike in all other experiments, for ProofWriter examples, we did not use the generic `spacy` named entity tagger because it did not support the entity types covered by ProofWriter examples. Instead, we used the real abstraction labels provided by the grammar files[2], and defined the following abstraction tokens: "*PERSON*", "*ATTRIBUTE*", "*ANIMAL*", "*RELATION*".

Table 3 shows the prediction accuracy on different slices of the D5 test set for all our models. Although trained on an older version of the dataset ("closed-world" assumption - CWA), we also report the original performance by Clark et al. (2020). While not directly comparable because of the different version of the dataset, we can still see that our `T5-small` experiments all extrapolate (D3-D5) better than the original `RoBERTa-large` model despite having less parameters. `RoBERTa-large` performs better at depths seen during training (D0-D2), which may be due to the model size being bigger (hence having greater capacity to model questions of previously seen depths), however, without abstraction, the larger model struggles to generalize to unseen reasoning depths, unlike our models.

Most importantly, if we compare with our "no abstraction" baseline model, Table 3 also shows that our abstraction methods help extrapolate to unseen reasoning depths. While our baseline model performs 83.7%, 77.3%, and 70% on examples from D3, D4, D5 respectively; all other abstraction-aware models extrapolate better, with the best overall model (`dec-loss`) performing 87.4%, 85%, and 80.6% on D3, D4, D5 examples respectively.

---

[2]https://tinyurl.com/proofwritergrammars

Table 4: Average test performance for all models on HotpotQA. Average and standard deviation computed with 3 random seeds.

| HOTPOTQA | ExactMatch | F1 | Precision | Recall |
|---|---|---|---|---|
| **no abstraction** | 54.7 ($\pm0.01$) | 68.9 ($\pm0.01$) | 72.4 ($\pm0.01$) | 69.1 ($\pm0.01$) |
| **emb-sum** | 54.3 ($\pm0.01$) | 68.4 ($\pm0.01$) | 72.0 ($\pm0.01$) | 68.6 ($\pm0.01$) |
| **emb-cat** | 52.6 ($\pm0.01$) | 66.5 ($\pm0.01$) | 69.7 ($\pm0.01$) | 67.0 ($\pm0.01$) |
| **enc-sum** | 54.2 ($\pm0.01$) | 68.5 ($\pm0.01$) | 72.0 ($\pm0.01$) | 68.7 ($\pm0.01$) |
| **enc-cat** | 52.9 ($\pm0.01$) | 67.2 ($\pm0.01$) | 71.0 ($\pm0.01$) | 67.3 ($\pm0.01$) |
| **dec-loss** | **55.7** ($\pm0.02$) | **69.8** ($\pm0.01$) | **73.3** ($\pm0.01$) | **69.9** ($\pm0.01$) |

We also note that, unlike in CLUTRR, in this dataset the depth of reasoning (D0-D5) is not tied to the input length. In ProofWriter, all examples have a similar average input length, regardless of the depth of reasoning required to predict the answer. Thus the `dec-loss` model is not penalized like it was the case in CLUTRR, which result in it being the best model overall.

Overall, we achieve new state-of-the-art results on the ProofWriter dataset when trained only on examples from D0-D2. This is suggesting that entity abstraction does help pre-trained Transformers to **compositionally generalize to unseen reasoning chains**. One important thing to note however is that both CLUTRR and ProofWriter sentences are relatively simple to abstract: 100% of the entities are correctly abstracted in both datasets. In the next sections we will see if explicit abstraction is still beneficial on more realistic but less formally defined tasks.

### 4.3 Multi-hop Question Answering with HotpotQA

In this section we report experiments on the multi-hop question answering (HotpotQA) dataset (Yang et al., 2018). HotpotQA contains natural language, making it more diverse and harder to get abstraction labels than CLUTRR. In addition, HotpotQA has 2-hop inference chains both in its training and testing data splits, thus we are not able to test generalisation to longer reasoning chains. Nevertheless, we believe it is a good compromise between natural language and multi-hop reasoning.

Used in the distractor setting, each example consists of a list of 10 Wikipedia paragraphs, a question that requires the model to combine information from two paragraphs, and the answer. Since concatenating all of the 10 paragraphs would result in a context size much larger than what regular Transformer models allowed (512 or 1024 tokens), we instead only took the two golden paragraphs as context, plus the question. While this beats the original purpose of retrieving the useful paragraphs, we are not interested in achieving state-of-the art on this benchmark. We are rather interested in using it to compare the usefulness of our approach on a more natural multi-hop question-answering setup. An example of input/output sequence pair can be seen in Appendix D.

Because the official test set is not public, we used the official validation set as our test set to compare our models and fine-tuned a `T5-small` model on 90% of the training set while keeping the remaining 10% as our custom validation set for early stopping. We trained one model for each of the 5 different strategies presented in Section 3, in addition to a baseline model fine-tuned without any abstraction knowledge.

Table 4 shows exact match, F1 scores, Precision and Recall on our test set. We can see from these results that the best of our model is the abstraction-aware `dec-loss` model (trained to predict both the answer and the input in its abstract form) with an F1 score of 69.8% against 68.9% for the baseline model ("no abstraction" row). However, the baseline model is a strong candidate and the abstraction does not always benefit the model depending on how it is incorporated. This may be due to the fact that entity abstraction labels are harder to predict on natural language, and that entity type abstraction may not be required in problems with little to no formal logical structure. We further discuss this in Section 6.

In an effort to contextualize these results, we note that our models performance is slightly above the "Query Focused Extractor" (Nishida et al., 2019) performance of 53.86 Exact Match and 68.06 F1. At the time of writing, the SOTA model on HotpotQA is the "From Easy to Hard" model (Li et al., 2022) with Exact

Table 5: Average test performance for all models on CoQA. Average and standard deviation computed with 3 random seeds.

| CoQA | ExactMatch | F1 |
|------|------------|-----|
| **no abstraction** | 66.0%(±0.001) | 74.4%(±0.000) |
| **emb-sum** | 65.7%(±0.002) | 74.2%(±0.002) |
| **emb-cat** | 63.8%(±0.001) | 72.3%(±0.001) |
| **enc-sum** | 65.8%(±0.001) | 74.1%(±0.002) |
| **enc-cat** | 65.2%(±0.002) | 73.8%(±0.002) |
| **dec-loss** | **66.4%**(±0.001) | **74.9%**(±0.001) |

Match of 71.89 and F1 score of 84.44. We note however that the test & training data in our setting is slightly different, so these are not perfectly comparable results. In addition, we are more interested in evaluating the effect of entity abstraction by comparing our model variants.

### 4.4 Conversational Question Answering with CoQA

Eventually, motivated by the use of a generative model, we test the same abstraction strategy in a conversational setting. For that, we leveraged the conversational question answering dataset CoQA (Reddy et al., 2019). The task presented by this dataset is to understand a text passage and answer a series of inter-connected questions in a conversation. The conversation aspect introduces follow-up questions that forces the model to keep track of what entity is currently being referred to and to look back at previous interactions. Examples of input/output sequence pairs can be found in Appendix D. The dataset does not always explicitly forces multi-hop reasoning steps, but it could still happen (*i.e.* see the last CoQA example of Appendix D in which the model must perform a substraction between all subjects and the ones already mentioned). In addition, the conversational nature of this dataset often introduces a co-reference step to be made before fetching the information from the paragraph in context. In this setting we will thus test if abstraction can help in this multi-step information retrieval procedure.

Similarly to HotpotQA, because the official test set is not public, we used the official validation set as our test set to compare our models and fine-tuned a `T5-small` model on 90% of the training set while keeping the remaining 10% as our custom validation set for early stopping. We trained one model for each of the 5 different strategies presented in Section 3, in addition to a baseline model fine-tuned without any abstraction knowledge.

Table 5 shows the exact match and F1 score on our test set. Although by a small margin, we can see that the best of our model is again the abstraction aware `dec-loss` model (trained to predict both the answer and the input in its abstract form) with an F1 score of 74.9% against 74.4% for the baseline ("no abstraction" row). The baseline model is quite strong already and the benefit of abstraction is questionable in this case. We further discuss this result in the following section. As in previous experiments, the worst performing model is `emb-cat` and one of the best is the `enc-sum` model (after `dec-loss` in this case).

In an effort to contextualize these results, we note that our models performance is better than DrQA + seq2seq with copy attention (Reddy et al., 2019) and BiDAF++ (Yatskar, 2018) with an F1 score of 67.0 and 71.6 respectively. The next closest model in the CoQA leaderboard is the FlowQA model (Huang et al., 2019) with an F1 score of 76.3. At the time of writing, the SOTA model on CoQA is RoBERTa+AT+KD (Ju et al., 2019) with an F1 score of 90.9. We note however that the test data in our setting is slightly different, so these are not perfectly comparable results. In addition, we are more interested in evaluating the effect of entity abstraction by comparing our model variants.

## 5    Discussion

In this section we aim to analyse further the results presented above by comparing them and measuring some key characteristic about each datasets.

Let's first summarize the results from all experiments. Overall, in all datasets (except for CLUTRR) the best model on average was the `dec-loss` model, which is trained to predict both the target output sequence and the input sequence in its abstracted form. As discussed previously, we believe that the reason why the `dec-loss` model did not perform as good as the other models in CLUTRR is because input sequence length is tied to the reasoning depth required to answer the question. Indeed, a question of level $n$ ($2 \leq n \leq 10$) will have exactly $n$ sentences in its input. Thus, during testing, the model must generate sequences of unseen lengths. Previous work showed that length generalization is a common weakness of classical language models (Murray & Chiang, 2018; Clark et al., 2020; Gontier et al., 2020; Anil et al., 2022; Press et al., 2022). Overall, our results suggest that **when input sequence length is relatively stable across all examples, the `dec-loss` abstraction strategy is beneficial to multi-step reasoning tasks**.

We then investigate why results on the two "natural", less procedural tasks (HotpotQA & CoQA) do not yield strong conclusions like in the synthetic cases of CLUTRR & ProofWriter. The first major difference between these datasets is that CLUTRR and ProofWriter are designed explicitly to test for compositional generalisation and reasoning extrapolation. In both datasets, the model is tested on examples of longer reasoning chains than what is observed during training. The language vocabulary is limited and the required reasoning depth is controlled. This is possible when working with tasks that are formally defined in a logical reasoning setting. On the other hand, HotpotQA and CoQA are designed from human written text (Wikipedia for Hotpot, human conversations for CoQA). This natural setting implies two things. (1) It is harder to control the reasoning depth required: all HotpotQA examples require to combine two pieces of information in order to answer the question. CoQA examples require to solve long co-reference resolution chains, however the test set does not contain longer chains of reasoning. (2) The language vocabulary is more noisy, making it harder to extract the useful information from a piece of text. These distinctions suggest that **entity type abstraction is mostly beneficial for formally defined logical tasks in which the model must reason at unseen depths during inference time**.

A second differentiating factor between these datasets is the quantity and quality of entity tags provided to the model. In an effort to estimate the influence of the tagger accuracy on our results, we compare each dataset in terms of the amount of entity tags they contain, as well as the accuracy of these tags. The reported performance of the entity tagger we used is 0.85 F1[3]. To further validate this metric, for each dataset, we estimate (i) the percent-

Table 6: Percentage of tokens being tagged as entities and estimated F1 score of the tagger on each dataset.

| Dataset | Entity Tokens | F1 |
|---|---|---|
| **CLUTRR** | 22.2% (±0.019) | 100% (±0) |
| **ProofWriter** | 36.0% (±0.036) | 100% (±0) |
| **HotpotQA** | 37.0% (±0.085) | 88.5% (±0.092) |
| **CoQA** | 17.7% (±0.095) | 88.4% (±0.085) |

age of tokens tagged as entities, (ii) the correctness of the tagged entities (precision), and (iii) the amount of correctly tagged entities out of all the entities that should have been tagged (recall). We estimate these metrics by manual inspection of a random set of examples for each dataset and report the F1 metric in Table 6. We can see that the percentage of tokens tagged as entities is roughly the same across datasets, except for CoQA which has fewer entity tokens. This is likely due to the fact that a lot of entities are referred to by co-reference in conversational QA. The average quality of entity tags for HotpotQA and CoQA is in line with the reported 85% F1 performance of the tagger. However, we note that all entities in CLUTRR and ProofWriter are correctly labeled. This distinction suggests that **the quality of the entity tagger can influence the benefits of performing entity type abstraction**.

To further estimate this influence, we ran experiments on the CLUTRR and ProofWriter benchmarks in which we added noise to the abstract labels, thus artificially simulating a weaker tagger. In particular, we modified 25%, 50%, and 75% of the entities by either replacing their entity type or by deleting the tag altogether. We report in Figure 1 the average answer accuracy of our models across extrapolation reasoning levels of the test set for both benchmark dataset (lvl 7-10 for CLUTRR and D3-D5 for ProofWriter). Overall, we can see that as the tagging noise increases, all models see their performance reduced and sometimes underperforming the baseline when the performance of the tagger is too weak. For instance once the noise reaches 50% or more, all models on ProofWriter and almost all models on CLUTRR (except `enc-sum` & `enc-cat`) become weaker than the baseline. This confirms that **as the tagger accuracy decreases, the overall**

---

[3]https://tinyurl.com/encoreweblg

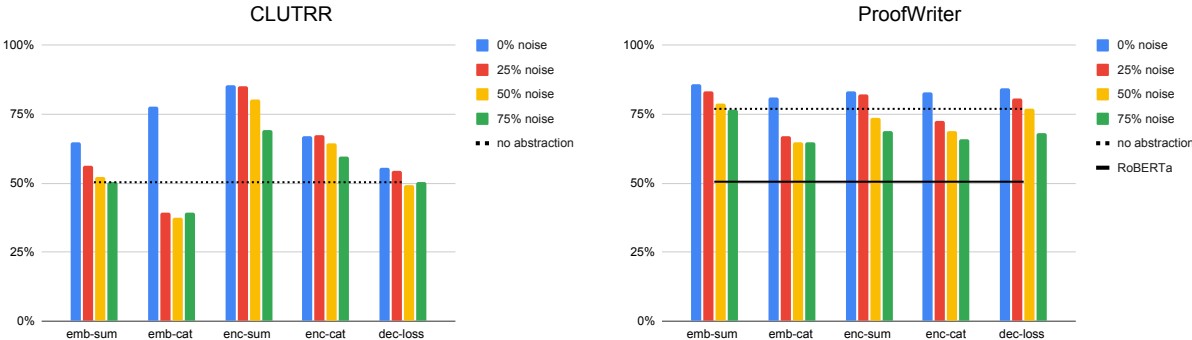

Figure 1: Average answer accuracy on extrapolation reasoning levels for models trained with 0%, 25%, 50% and 75% entity tagger noise on CLUTRR and ProofWriter. The baseline model (no abstraction) is represented by a black dotted line. Previous work on ProofWriter is represented by a black solid line.

**performance of the model does so to**. In addition, it also shows that it is better for the model to not have additional complications (abstraction in this case) if the additional parameters needed for that are used on noise more than 50% of the time.

In an effort to better characterize the failure cases of our models on the more natural datasets we report in Appendix F model predictions on various datasets. In particular, Appendix F.2 shows examples of HotpotQA in which abstraction models answered correctly but not the baseline (no-abstraction) model and vice-versa. Overall, we observe that many examples in HotpotQA don't require multiple reasoning hops, which can be an additional source of noise beside the tagging accuracy.

Overall, we believe three factors are influencing our results on the more natural tasks: (i) the effect of a weaker entity tagger performance as shown in Table 6 and Figure 1, (ii) the depth of reasoning being relatively shallow and the same across training and testing sets as observed in Appendix D and F.2, (iii) the natural and conversational language being further away from a formal language. These factors result in "real-world" problems not having strong enough logical structure which can benefit from the abstraction technique. It is only when tasked on more logical problems explicitly requiring reasoning depths unseen during training that abstraction becomes significantly beneficial.

## 6  Conclusion

**Conclusion**. We presented various ways to incorporate abstract knowledge into Transformer Language Models. Focusing on entity types, this work evaluated model performance on reasoning tasks requiring compositional generalization and multi-hop reasoning. Overall our results demonstrate three things: (i) incorporating abstract knowledge significantly improves reasoning and compositional generalization in both interpolation and extrapolation when the environment is formally defined in a logical reasoning setting; (ii) different ways to incorporate abstraction yields different performance boosts: `enc-sum` and `dec-loss` are generally performing better than others; (iii) abstraction is not beneficial when the task at hand is more natural, less procedural, and not requiring long reasoning chains. This last result is due to the noisy entity tagging from "off-the-shelf" taggers, and due to the nature of the task at hand.

**Limitations**. One limitation of our work is that the method we present requires annotated data which is not always available. Furthermore, this additional data processing can take time and may not scale well to larger datasets if implemented naively. Overall, we did not find significantly longer training time for all approaches except for the `dec-loss` model that was training on average 2 times longer than the other methods. The most important factor in training time was the dataset used rather than the method used. Models trained on natural language datasets CoQA and HotpotQA took days to train while models trained on CLUTRR and ProofWriter took a few hours.

**Future Explorations**. One hypothesis that results from our work is the following question: could *pre-training* Transformer models with additional abstraction data result in stronger performance when fine-tuned with abstraction data like we do in this work? Although we could not train *from-scratch* a T5 model on its original C4 dataset (Raffel et al., 2020), we believe that augmenting C4 with abstract annotations like we do on a smaller scale and training T5 from scratch on this augmented dataset could potentially yield a stronger language model.

Another interesting future direction worth exploring would be the ethical benefit of incorporating explicit abstraction into large language models. Previous work showed that pre-trained language models can have some undesired societal biases (Henderson et al., 2018; Shwartz et al., 2020; Bender et al., 2021). Although not explored in this work, we believe that giving abstract entity types like we do in this work could have a positive societal impact on language models, potentially alleviating some of these biases. For instance, through abstraction, a model can be exposed to female and male names both being "*PERSON*"s and that a "*PERSON*" can equally be a "*manager*" or an "*assistant*" regardless of its gender. Similar exposure could be achieved with other abstraction types such as "*RELIGION*", "*JOB*", "*COUNTRY*", "*NATIONALITY*", etc... Benchmarks such as StereoSet (Nadeem et al., 2021) could be used to measure the beneficial impact that explicit abstraction can have.

### Acknowledgments

The authors acknowledge support from ServiceNow Research for providing computational resources which were used to run the experiments in this work. The first author is partially funded by a scholarship from the Fonds de Recherche du Quebec Nature et Technologie (FRQNT). We thank CIFAR for their support through the CIFAR AI Chairs program. We also thank NSERC and PROMPT for their support.

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

## A  Hyperparameters

We used the default `T5-small` hyperparameters from the `HuggingFace` library (Wolf et al., 2019). We present in Table 7 below the library version we used and the model hyperparameters used for all experiments.

The only difference between each dataset is the number of same-type entity tags allowed in each input sequence $n$. These values were chosen to be the smallest possible number while still being able to identify all the different entities in one sequence. For CLUTRR the maximum number of same-type entities was 20, for ProofWriter it was 10, and for HotpotQA and CoQA it was 100. This is due to the larger context size and the diversity of natural language texts. Results on CLUTRR when $n = 1$ can be found below in Appendix C.

| | |
|---:|:---:|
| $n$ for CLUTRR | 20 |
| $n$ for ProofWriter | 10 |
| $n$ for HotpotQA | 100 |
| $n$ for CoQA | 100 |
| AllenNLP | version 2.2.0 |
| Transformers | version 4.4.2 |
| Spacy | version 2.3.5 |
| batch size | 256 |
| 16-bit floating point | True |
| dim embedding | 512 |
| dim feedforward | 2048 |
| dim key-value | 64 |
| dropout | 0.1 |
| max length | 512 |
| #of heads | 8 |
| #of layers | 6 |
| optimizer | AdamW |
| learning rate | 1.00E-05 |
| betas | [0.9, 0.999] |
| epsilon | 1.00E-08 |
| gradient norm | 1.0 |
| sampler | top-p |
| p | 0.9 |
| temperature | 1.0 |

Table 7: Library version and model hyper-parameters.

## B  On the abstraction accuracy of the `dec-loss` model

After training the `dec-loss` models for each task, one question we might ask is whether the model is able to correctly predict abstract tokens with its dedicated language model head.

After generating abstract sequences for all test sets, we measured that the model is correctly predicting entity *types* 100% of the time for CLUTRR and ProofWriter, and around 70% of the time for HotpotQA and CoQA compared to the types predicted by our "off-the-shelve" tagger.

One interesting finding though, is that the model is not consistent with entity numbers across the same example. While it can correctly predict the *type* of an entity ("PERSON" vs "LOCATION"), it is almost impossible to stay consistent with entity IDs of that type ("PERSON_11" vs "PERSON_23"). This suggests that at inference time, the distribution of abstract tokens belonging to the same entity type is very close to uniform, which also suggests that all these token embeddings are close to each other.

## C   CLUTRR results when $n = 1$

To better understand the effect of the hyper-parameter $n$ in our models, we ran additional experiments on the CLUTRR benchmark for different values of $n$. When $n = 20$ all entities in each example gets a unique abstract token ID. When $n = 1$, all entities of the same type are mapped to the same abstract token. In order to keep track of which entity is related to which, token identity must be preserved. When $n = 1$, the only way to preserve token identity is for the model to use the original (non abstracted) sequence. We report below the average answer accuracy of our models on CLUTRR for different values of $n$. The average is taken across all test levels from 2 to 10.

|                 | n = 20    | n = 1    |
|-----------------|-----------|----------|
| no abstraction  | 62.9%     |          |
| emb-sum         | 71.5%     | **75.0%** |
| emb-cat         | **64.8%** | 30.4%    |
| enc-sum         | **88.8%** | 85.3%    |
| enc-cat         | **73.8%** | 36.1%    |
| dec-loss        | 64.2%     | **75.7%** |

Table 8: Average CLUTRR answer accuracy for models trained with $n = 20$ and $n = 1$ token per entity tag.

We can see that the `emb-cat` and `enc-cat` models perform much worse in the $n = 1$ setting, while the other models are relatively similar in both settings. This suggests that concatenating representations together and then performing a matrix multiplication to resize the representation does not keep entity identity as well as the other methods.

## D  Input - Output examples

**CLUTRR**.

| | |
|---|---|
| lvl.2 input: | "question :  How is Anne related to Gary ?  context :
Brett is Anne 's father .  Gary is a son to Brett ." |
| output: | "answer :  Anne has a brother named Gary ." |
| lvl.4 input: | "question:  What is the family connection between Patricia
and Timothy ?  context :  May is the aunt of Doris .  Patricia
has a daughter called Doris .  Timothy is Charles 's brother ."
May has a son called Charles . |
| output: | "answer :  Timothy is the nephew of Patricia ." |

**ProofWriter**.

| | |
|---|---|
| Depth-0 input: | "context :  The cow is round.  The cow needs the lion.  The cow
needs the rabbit.  The cow sees the lion.  The cow visits the rabbit.
The lion is round.  The rabbit is kind.  The rabbit visits the tiger.
The tiger is big.  The tiger is kind.  The tiger sees the rabbit.
The tiger visits the rabbit.  If something is kind and it visits the
rabbit then it is young.  If something sees the tiger and it visits
the lion then it sees the rabbit.  If something is big and young then
it sees the lion.  If something visits the rabbit then the rabbit
needs the lion.  If something is big then it visits the rabbit.  If
something sees the tiger then it is rough.  If something visits the
rabbit and it is kind then the rabbit needs the lion.  If something
is rough and kind then it visits the lion.  If something needs the
lion then it is big.  question :  The tiger visits the rabbit." |
| Depth-0 output: | "answer :  True" |
| Depth-2 input: | "context :  Anne is nice.  Charlie is blue.  Charlie is furry.
Charlie is green.  Charlie is kind.  Charlie is nice.  Charlie is
red.  Fiona is furry.  Fiona is green.  Harry is furry.  Harry is
kind.  Harry is nice.  All nice people are rough.  If someone is
red and furry then they are blue.  Rough, kind people are furry.
If Charlie is furry then Charlie is nice.  All furry people are
nice.  If someone is rough then they are kind.  If someone is red
and nice then they are blue.  All furry people are red.
question :  Harry is not blue." |
| Depth-2 output: | "answer :  False" |

**HotpotQA**.

| | |
|---|---|
| input: | "question :  Which magazine was started first Arthur's Magazine or
First for Women ?  context :  Arthur's Magazine (1844-1846) was an
American literary periodical published in Philadelphia in the 19th
century.  Edited by T.S. Arthur, it featured work by Edgar A. Poe,
J.H. Ingraham, Sarah Josepha Hale, and others.  In May 1846 it was
merged into "Godey's Lady's Book".  First for Women is a woman's
magazine published by Bauer Media Group in the USA. The magazine
was started in 1989.  It is based in Englewood Cliffs, New Jersey.
In 2011 the circulation of the magazine was 1,310,696 copies." |
| output: | "answer :  Arthur's Magazine" |

**CoQA**.

| | |
|---|---|
| context: | "context :  The Vatican Apostolic Library, more commonly called the Vatican Library or simply the Vat, is the library of the Holy See, located in Vatican City.  Formally established in 1475, although it is much older, it is one of the oldest libraries in the world and contains one of the most significant collections of historical texts.  It has 75,000 codices from throughout history, as well as 1.1 million printed books, which include some 8,500 incunabula.  The Vatican Library is a research library for history, law, philosophy, science and theology.  The Vatican Library is open to anyone who can document their qualifications and research needs."  [...]  "Only a handful of volumes survive from this period, though some are very significant." |
| input: | context above + "question :  When was the Vat formally opened?" |
| output: | "answer :  It was formally established in 1475." |
| input: | context above + "question :  When was the Vat formally opened?  answer : It was formally established in 1475.  question :  what is the library for?  answer :  research.  question :  for what subjects?" |
| output: | "answer :  history, and law." |
| input: | context above + "question :  When was the Vat formally opened?  answer : It was formally established in 1475.  question :  what is the library for?  answer :  research.  question :  for what subjects?  answer : history, and law.  question :  what else ?" |
| output: | "philosophy, science and theology." |

## E    GeoQuery Results

In this section we report results of our different models on the GeoQuery (Zelle & Mooney, 1996) benchmark. In particular, we use two versions described by Shaw et al. (2021), focusing on compositional generalization: a train/test split based on query length and a train/test split based on Target Maximum Compound Divergence (TMCD). Each train and test set contains 440 examples.

We trained our models on both of these splits and evaluated the exact match of the predicted output of our models. Below are our average results based on 3 random seeds:

|  | **LENGTH split** | **TMCD split** |
|---|---|---|
| **no-abstraction** | 24.24% (+- 0.001) | **36.06%** (+- 0.001) |
| **emb-sum** | **27.05%** (+- 0.000) | 34.47% (+- 0.001) |
| **emb-cat** | 15.91% (+- 0.002) | 18.56% (+- 0.001) |
| **enc-sum** | 18.64% (+- 0.002) | 35.08% (+- 0.001) |
| **enc-cat** | 17.50% (+- 0.000) | 27.58% (+- 0.001) |
| **dec-loss** | 24.09% (+- 0.000) | 30.38% (+- 0.001) |

We can see that for the length split, one abstraction model performs better than the baseline (no-abstraction), but on the TMCD split, models with abstraction are not better than the baseline. However, we believe this dataset may not be a good candidate to showcase if abstraction is useful for 2 reasons:

First, all examples in the dataset have a few number of entities (2 maximum):

| entities per example | examples | percentage of the dataset |
|---|---|---|
| 0 | 207 | 23.5% |
| 1 | 667 | 75.8% |
| 2 | 6 | 0.7% |
| **Total** | **880** | **100%** |

This makes each abstract input very similar to the original input. Thus the abstract sequence provides very little information to the model.

Second, the dataset as a whole contains a few number of entities. 12% of tokens per input are tagged as entities. This is a relatively small number compared to the other datasets as we can see below.

Fraction of tokens per input in each dataset identified as entities by spacy NER:

| **Data** | **avg (+-std)** |
|---|---|
| CLUTRR | 0.22 (+- 0.019) |
| Proofwriter | 0.36 (+- 0.036) |
| HotpotQA | 0.37 (+- 0.085) |
| CoQA | 0.18 (+- 0.095) |
| Geoquery | 0.12 (+- 0.080) |

We can say from these results that in order for the abstraction technique to be useful, the dataset must contain at least more than 2 entities per input sequence.

# F Prediction examples

## F.1 CLUTRR

In this section we present output examples of our models on the CLUTRR level 3 test set.

Table 9: CLUTRR test set level 3 output examples. Correct answers are in green and wrong answers are in red for better visibility.

| input | no abstraction | emb-sum | emb-cat | enc-sum | enc-cat | dec-loss |
|---|---|---|---|---|---|---|
| Jonathan is the father of Anne . Anne is a aunt to Stephanie . Stephanie is the daughter of Veronica . How is Veronica related to Jonathan ? | Jonathan is Veronica 's father | Jonathan is Veronica 's father | Jonathan is Veronica 's uncle | Jonathan is Veronica 's father | Jonathan is Veronica 's father | Jonathan is Veronica 's father |
| Stephanie is the granddaughter of Jonathan . Bryant is Stephanie 's father . Eric is a son of Bryant . How are Jonathan and Eric related to each other ? | Eric is the grandson of Jonathan | Eric is Jonathan 's grand-son | Eric is Jonathan 's grand-son | Eric is Jonathan 's grand-son | Eric is Jonathan 's grand-son | Eric is Jonathan 's grand-son |
| Joyce has a daughter called Stephanie . Betty is a grandmother to Stephanie . Paul has a wife who is Joyce . For Paul , who is Betty ? | Betty is Paul 's mother-in-law | Betty is Paul 's mother-in-law | Betty is Paul 's aunt | Betty is Paul 's mother-in-law | Betty is Paul 's mother-in-law | Betty is Paul 's mother-in-law |
| Bryant is a son of Jonathan . Jonathan is the husband of Betty . Craig is a son of Bryant . Who is Craig from the point of relation of Betty ? | Craig is Betty 's son | Betty has a grandson who is Craig | Craig is Betty 's grandson | Betty has a grandson who is Craig | Craig is Betty 's son | Craig is the son-in-law of Betty |
| Stephanie is a granddaughter to Betty . Anne is Bryant 's sister . Bryant is a brother of Stephanie . Who is Anne from the point of relation of Betty ? | Anne is Betty 's grand-daughter | Anne is Betty 's grand-daughter | Anne is Betty 's grand-daughter | Anne is Betty 's grand-daughter | Anne is Betty 's grand-daughter | Anne is Betty 's grand-daughter |
| **score** | **4/5** | **5/5** | **3/5** | **5/5** | **4/5** | **4/5** |
| score on lvl.3 test set reported in Table 2 | 84.4% | 85.3% | 60.0% | 94.8% | 86.1% | 74.7% |

Out of these 5 examples, the `emb-cat` model makes 2 mistakes (line1 & line3), the `no-abstraction`, `enc-cat`, `dec-loss` models make 1 mistake (line4) and the `emb-sum` and `enc-sum` models make 0 mistake.

The `emb-cat` model mistakes (line1 & line3) are due to the prediction of uncle/aunt relations instead of father/mother-in-law respectively. It is interesting to note that these relations are similar in the sense that they all link one generation to the one just above.

It is also interesting to note that the example in which the `no-abstraction`, `enc-cat` and `dec-loss` models fail (line4), the `emb-cat` model on the other hand answers correctly. In this specific example the correct relationship is "grandson" which links one generation to 2 generations below, however the mistakes made by the failing models only link one generation to 1 below (son, son, son-in-law).

## F.2   HotpotQA

In this section we present HotpotQA examples in which all our abstraction models correctly answered the question but not the baseline (no abstraction) model.

```
--------------------
Question: Are both Elko Regional Airport and Gerald R. Ford International Airport
          located in Michigan?
facts   : [
    "Elko Regional Airport (IATA: EKO, ICAO: KEKO, FAA LID: EKO) , formerly Elko
     Municipal Airport, is a mile west of downtown Elko, in Elko County, Nevada.",
    "Gerald R. Ford International Airport (IATA: GRR, ICAO: KGRR, FAA LID: GRR)
     is a commercial airport in Cascade Township approximately 13 mi southeast
     of Grand Rapids, Michigan. The facility is owned by the Kent County Board of
     Commissioners and managed by an independent authority. The Federal
     Aviation Administration (FAA) National Plan of Integrated Airport Systems for
     2017-2021 categorized it as a small hub primary commercial service facility."]
Answer  : no
baseline: yes
embsum  : no
embcat  : no
encsum  : no
enccat  : no
decloss : no
--------------------
```

This is a yes/no question with many acronyms and entities. The abstraction can be beneficial here in order to simplify the context.

```
--------------------
Question: What act for Innocent Records achieved Platinum sales and shares its name
          with a primary color in the RGB color model?
facts   : [
    "Blue is the colour between violet and green on the spectrum of visible light.
     Human eyes perceive blue when observing light with a wavelength between 450 and
     495 nanometres. Blues with a higher frequency and thus a shorter wavelength appear
     more violet, while those with a lower frequency and a longer wavelength gradually
     appear more green. Pure blue, in the middle, has a wavelength of 470 nanometres.
     In painting and traditional colour theory, blue is one of the three primary colours
     of pigments, along with red and yellow, which can be mixed to form a wide gamut
     of colours. Red and blue mixed together form violet, blue and yellow together
     form green. Blue is also a primary colour in the RGB colour model, used to create
     all the colours on the screen of a television or computer monitor.",
    "Innocent Records was a pop record label created to cater to for EMI's Virgin Records
     more pop oriented acts. Following the success of the Spice Girls, Virgin Records
     decided to delve into the pop market. In doing so they poached Hugh Goldsmith from
     RCA Records (famous for steering Take That's initial flagging sales, to a multi-
     platinum act). They let him launch his own Virgin Records offshoot. His first
     signing was Billie Piper, followed by Martine McCutcheon, along with several dance
     acts Todd Terry to name one. The label continued to thrive well into the mid-2000s
     with Atomic Kitten and Blue achieving Platinum sales."]
Answer  : Blue
baseline: Atomic Kitten
embsum  : Blue
embcat  : Blue
```

```
encsum  : Blue
enccat  : Blue
decloss : Blue
-------------------
```

This is a question with long paragraphs in the context, abstracting entities could help simplify the context to better answer the question. Nevertheless, note that if we assume the model already knows that blue is a color (or that Atomic Kitten is not a color), then the second paragraph is enough to answer the question.

```
-------------------
Question:  William Cammisano was part of which Mafia family?
facts   : [
    "William "Willie Rat" Dominick Cammisano Sr. (April 26, 1914 - January 26, 1995) was
     a Kansas City, Missouri, mobster and enforcer for Nicholas Civella\'s Kansas
     City crime family.",
    "The Kansas City Crime Family, also known as Civella crime family (pronounced ] ),
     is a Mafia family based in Kansas City, Missouri."]
Answer  : Kansas City crime family
baseline: Nicholas Civella
embsum  : Kansas City Crime Family
embcat  : Kansas City Crime Family
encsum  : Kansas City Crime Family
enccat  : Kansas City Crime Family
decloss : Kansas City Crime Family
-------------------
```

Here the baseline model predicted the owner of the mafia family, which is also used to refer to the mafia group according to the second paragraph. Note again, the question can be answered with the first paragraph.

```
-------------------
Question: Who is older out of Bob Saget, the American comedian, and Indian director
          S. Shankar?
facts   : [
    "Shankar Shanmugam (born 17 August 1963), credited mononymously as Shankar, is an
     Indian film director and producer who predominantly works in Tamil cinema. He was
     identified by S. A. Chandrasekhar. Recognized for directing high budget films, he
     is also a pioneer of vigilante movies in Tamil. He made his directorial debut in
     "Gentleman" (1993) produced by K. T. Kunjumon, for which he was awarded the Filmfare
     Best Director Award and the Tamil Nadu State Film Award for Best Director. He is the
     highest paid film-maker in India among his contemporaries.",
    "Robert Lane "Bob" Saget (born May 17, 1956) is an American stand-up comedian, actor,
     and television host. His television roles include Danny Tanner on the ABC sitcom
     "Full House" (1987-95) and its Netflix sequel "Fuller House", and hosting
     "America\'s Funniest Home Videos" from 1989 to 1997. Saget is also known for his
     adult-oriented stand-up routine. He also provided the voice of the future Ted Mosby
     on the CBS sitcom "How I Met Your Mother" from 2005 to 2014."]
Answer  : Robert Lane "Bob" Saget
baseline: Ted Mosby
embsum  : Robert Lane
embcat  : Robert Lane " Bob " Saget
encsum  : Robert Lane
enccat  : Robert Lane " Bob " Saget
decloss : Robert Lane " Bob " Saget
-------------------
```

This question asks to compare two different entities of the same type. The baseline model answers with a person's name but not the correct one. Since there are lots of entities in this example, abstraction can be beneficial to simplify the context and select the correct person in the end.

Next, we present HotpotQA examples in which all our abstraction models incorrectly answered the question but not the baseline (no abstraction) model.

```
--------------------
Question: What city does Paul Clyne and David Soares have in common?
facts   : [
    "Paul Clyne was the District Attorney of Albany County, New York from January 2001
     through December 2004. A graduate of Albany Law School, he spent about 14 years as
     an assistant district attorney, before he was tapped by local politicians to replace
     the retiring District Attorney, Sol Greenberg. He was defeated for re-election by
     David Soares, first in the Democratic Party primary election in September 2004, and
     then in the general election in November 2004, in which he ran on an independent
     line. After a stint teaching at the New York Prosecutors Institute, he went into
     private practice as a criminal defense attorney in 2007, with an office in Albany,
     New York.",
    "P. David Soares (born October 26, 1969, Brava, Cape Verde) is the Albany County,
     N.Y. District Attorney. He is a Democrat."]
Answer  : New York
baseline: New York
embsum  : Albany
embcat  : Albany
encsum  : Albany
enccat  : Albany
decloss : Albany
--------------------
```

In this example the abstraction models all answered 'Albany', which is also a valid answer. In fact, the question asks for a city and not a state. In addition, the second paragraph doesn't mention New York City, so Albany should probably have been the correct answer.

```
--------------------
Question: what producer of The Real Housewives of Orange County also hosts
          "Watch What Happens Live with Andy Cohen"?
facts    : [
    "Andrew Joseph "Andy" Cohen (born June 2, 1968) is an American talk show and radio
     host, author and producer. Cohen hosts the Bravo nightly series "Watch What Happens
     Live with Andy Cohen". He is the first openly gay host of an American late-night
     talk show. After being head of development at Bravo for more than 10 years, Cohen
     resigned in November 2013. He continues to serve as an executive producer of "The
     Real Housewives" franchise.",
    "The eleventh season of "The Real Housewives of Orange County", an American reality
     television series, is broadcast on Bravo. It aired June 20, 2016, until November 21,
     2016, and is primarily filmed in Orange County, California. Its executive producers
     are Adam Karpel, Alex Baskin, Douglas Ross, Gregory Stewart, Scott Dunlop,
     Stephanie Boyriven and Andy Cohen."]
Answer   : Andy Cohen
baseline: Andy Cohen
embsum   : Adam Karpel
embcat   : Adam Karpel
encsum   : Adam Karpel
enccat   : Adam Karpel
```

```
decloss : Adam Cohen
--------------------
```

In this example the abstraction models all answer with the first entity listed as being a producer of The Real Housewives, eventhough it is not the person who hosts What Happens Live. Note that this example contains the answer in the question, which may be confusing the abstraction models.

```
--------------------
Question: What Actor whose birth name was Charles Dennis Buchinsky, was part of the
          Leslie Nielsen comedy?
facts   : [
    "Charles Bronson (born Charles Dennis Buchinsky; Lithuanian: "Karolis Dionyzas
     Bučinskis" ; November 3, 1921 - August 30, 2003) was an American actor.",
    "Allan A. Goldstein (born May 23, 1949) is an American film director and
    screenwriter, perhaps best known for directing the Charles Bronson vehicle and
    the Leslie Nielsen comedy".]
Answer  : Charles Bronson
baseline: Charles Bronson
embsum  : Allan A. Goldstein
embcat  : Allan A. Goldstein
encsum  : Allan A. Goldstein
enccat  : Allan A. Goldstein
decloss : Allan A. Goldstein
--------------------
```

In this example the answer can be answered directly from the first paragraph since we are asking for the name of the actor born 'Charles Denis Buchinsky'. Here the abstraction models all answered with the person from the second paragraph. This is probably due to the fact that a two hop question, starting with entity 'Charles Dennis Buchinsky', and asking for a person would first link the first and second paragraph with the entity 'Charles Bronson', and then answer with the entity 'Allan A. Goldstein'.

