# OpenReview forum: "Does Entity Abstraction Help Generative Transformers Reason?"
_TMLR — Accepted by TMLR_

### Review · Reviewer_M3sQ · 2022-07-11

**Summary Of Contributions:**

The paper focuses on presenting a new architecture for incorporating abstract entity types into NLP tasks. The paper aims for a general method, covering four NLP tasks (relational reasoning, adductive reasoning, multi-hop QA, conversational QA). They try multiple model variants for injecting entity abstractions. The results are somewhat mixed — while it shows improvements in more synthetic datasets, the gain almost dissppears when moving to the natural language dataset.

**Broader Impact Concerns:**

I do not have broader impact concerns with this work.

**Requested Changes:**

See the weaknesses section above. Overall, I think the paper is pretty weak in its current form.

**Strengths And Weaknesses:**

Strength: The motivation for this work is solid, and the paper is overall written clearly such that it is easy to follow.

Weaknesses:
I see little novelty in the method. All architectures are fairly standard, embedding features and concatenating them, and summing them in multiple configurations.

I'm unconvinced why this is about "abstraction" as they are learning a unique embedding for each entity, similar to the entity as experts (Fevry et al, EMNLP 2022) paper?  -- this was my misunderstanding, thanks for the clarification!

The pipelined nature of the proposed model poses a weakness — as it relies on the accuracy of entity tagger (which should be reported and understood in the domains they study). More generally, if the entity tagger was able to tag the entity means the information is present in the original text, which end-task application should also be able to exploit them directly?

The experiments on Proofwriter seems broken to me — as they are using metadata (entity types from the grammar file which is _used_ to generate the data). It’s hard to claim state-of-the-art results in this setting, where their model is using more information than other baseline methods.

The entity types that are considered in this work are fairly coarse. I hope they provide at least a few motivating examples where such coarse-grained entity types can be useful for end tasks. It might be interesting to see if more fine-grained entity types can be helpful.

The entity indexing makes me question what happens to entities that are newly arising at test time. I'm also unclear how much of this is "abstraction", if the model is learning a separate embedding for each unique entities? I also find it hard to find that the number of same-type entities in hotpotQA and coQA is only 100..?? What is the dataset it is using exactly?

The results should be in context with existing work, not just a baseline of its own. Except for Table 3 (experiments on Proofwriter), all other results are not in context with prior work, which makes it harder to evaluate this proposed method’s effectiveness.

We should also compare the model capacity across different configurations, as far as I understand, the proposed model have increased parameter counts and that is potentially why it outperforms baseline.

I’m not sure the gains in Table 4 are meaningful — some study on variation will be necessary.

The basic statistics such as the percentage of tokens tagged as entities, length of entity spans, etc, should be reported to understand the results better (esp. for natural dataset, HotpotQA and CoQA). More analysis should follow as well — for example, how does the entity tagger performance affect the performance of the final model?

---

> ### Author Response · Authors · 2022-07-20
> **initial response**
>
> Thank you for taking the time to read our paper and giving us your feedback.
>
> **About architecture novelty**:
>
> Indeed, we do not propose a novel architecture as that is not the goal of this work. We perform an analytical *study on the effect of entity types* for pre-trained transformer models on various tasks.
>
> **About abstraction and learning “*a unique embedding for each entity*”**:
>
> We do not learn a unique embedding for each entity.
> We learn $n$ different embeddings for each entity type with $n$ being much much smaller than the total number of entities in the dataset. For instance in CLUTRR $n=20$ but there exists $135$ different unique entities of the same type. The only reason we have $n > 1$ is that we believe that for some datasets it is important to differentiate between two entities of the same type within the same sequence. We updated the beginning of Section 3 to try to clarify this important aspect of our methodology. In addition, note that in order to better understand the influence of having multiple tags for the same entity type, we also ran experiments in which we set $n=1$, thus forcing all entities of the same type to be mapped to the same embedding. Results can be seen in Appendix C.
>
> **About relying on a fixed pre-trained entity tagger**:
>
> We briefly discuss this in the discussion section of our paper. The reported performance of the entity tagger we used is 0.85 F1. Furthermore, to estimate the impact of the tagger accuracy, we ran experiments on the CLUTRR benchmark in which we added 10% noise to the abstract labels, thus simulating a weaker tagger. Results (in Appendix D) show greater variance among the models we use. The stronger model without noise (enc-sum) is still the best model with noise. It is interesting to see that three models (emb-sum, enc-sum, dec-loss) still beat the baseline and do not lose much performance or perform slightly better with noise. On the other hand, weaker models like emb-cat or dec-cat see their performance greatly reduced and are under-performing the baseline. This suggests that the tagger accuracy can play a role in the overall performance. However this also shows that some methods are much more robust against weaker tagging accuracy.
>
> **Short answers**:
>
> > if the entity tagger was able to tag the entity means the information is present in the original text, which end-task application should also be able to exploit them directly?
>
> We are not sure to understand this question, but overall we believe that models required to generalize to unseen lengths of reasoning chains can benefit from entity type abstraction.
>
> > motivating examples where such coarse-grained entity types can be useful
>
> Results in Tables 2 & 3 show that such abstraction significantly improves compositional generalization to unseen lengths of reasoning chains in multi-step reasoning tasks.
>
> > what happens to entities that are newly arising at test time.
>
> New entities will be abstracted by the pre-trained tagger in its best possible way. In addition, as mentioned previously, three of the five methods are robust to an additional 10% noise on top of the current tagger capacity.
>
> > how much of this is "abstraction", if the model is learning a separate embedding for each unique entities?
>
> As mentioned above, we do not learn a separate embedding for each unique entity.
>
> > I also find it hard to find that the number of same-type entities in hotpotQA and coQA is only 100..??
>
> Exactly, it is not 100. It is much more. 100 is the number of abstract tokens we create for each entity **type**.
>
> > results are not in context with prior work
>
> We use the datasets in ways that prior work did not, making it harder to compare against. Nevertheless, our goal is not to achieve state-of-the-art on various tasks, but rather understand when it is beneficial to explicitly provide entity type abstraction to pre-trained transformer models. Our method is flexible enough that it can benefit a wide range of models.
>
> > I’m not sure the gains in Table 4 are meaningful
>
> Indeed, the gains are not significant, and this is part of our conclusion: “*abstraction is not beneficial when the task at hand is more natural, less procedural, and not requiring long reasoning chains*”. Our goal is not to propose a new state-of-the-art method but to highlight when entity type abstractions can be helpful and when they are not.
>
> > More analysis should follow as well — for example, how does the entity tagger performance affect the performance of the final model?
>
> As mentioned above and in Appendix D, the performance of the entity tagger has an effect on two of the 5 different methods we present.
>
> We hope these clarifications will improve our paper and your understanding. We will add additional statistics (tokens tagged as entities, comparison against model with extra parameters) in the next revisions, until then don’t hesitate to continue the discussion on the above points if you have any questions.

---

> ### Author Response · Authors · 2022-08-08
> **Some statistics about tagged entities**
>
> Hi,
>
> Below are the number of tokens being tagged by the entity tagger per input sequence for each dataset used.
>
> | Data | avg (+-std) |
> | --- | --- |
> | CLUTRR | 0.23 (+- 0.043)
> | Proofwriter | 0.32 (+- 0.002)
> | HotpotQA | 0.37 (+- 0.085)
> | CoQA | 0.18 (+- 0.095)
>
> We can see that it is roughly the same across datasets.
> We will continue with a manual analysis on the quality of these tagged labels and will add the final table in the paper.

---

### Review · Reviewer_7k7T · 2022-07-11

**Summary Of Contributions:**

* The paper uses several methods to integrate information about entity types for four tasks, focusing on compositional generalization and reasoning – two are synthetic tasks and two are non-synthetic.

* The methods are very similar to ones used to integrate different types of annotations in combination with pre-trained LMs before, for example entity linking annotations or pos tags/syntax.

* There are substantial improvements on two synthetic tasks – CLUTRR and ProofWriter – requiring reasoning and compositional generalization. There are small to no improvements on the real NLP tasks of HotPotQA and ConQA, which also require some reasoning.


**Broader Impact Concerns:**

No concerns.

**Requested Changes:**

* I suggest evaluation on a non-synthetic task of compositional generalization, such as semantic parsing on GeoQuery, also without providing to the system entity types derived from the logical form (without giving extra oracle information), or non-synthetic tasks of solving math word problems that require multiple steps of reasoning

* I suggest discussing the computational cost of the methods to integrate entity types

* I suggest addressing the clarity concerns outlined in the previous section

* (minor) P.2 “ … Observe that some mehtods are more efficient than others” → do you mean effective?

* (minor) Sec 3, first para: “Use existing tools such as spacy” → if you used spacy please say directly “We used the spacy named entity tagger …” Otherwise one may think you used a different tool that is similar to spacy.




**Strengths And Weaknesses:**

* The paper is generally well written. There is a lot of detail on the methods and results (but some details are missing or presented a bit late, see weaknesses)

* The research question is interesting, and the comparison of many different methods to use entity type information is interesting.

* It is good that experiments on natural data were included in addition to ones on synthetic data. The authors have been very honest about the results.

* Clarity: the methods to integrate the entity types were not entirely clear: initially I thought there are some intermediate pre-training tasks on unlabeled text, but I think what is done end-task finetuning while integrating entity type information in various ways – as features, or as an auxiliary prediction task.

* Clarity:  In section 3.3 Abstraction as auxiliary task, if this is done at task fine-tuning time, Y and X_s will have different lengths. I believe the equation should be updated to indicate sums /averages of per-position cross-entropies , including introducing notation for the lengths.

* Datasets used in experiments: there are some more natural datasets that have been the standard for compositional generalization evaluations, such as ones based on compositional splits of GeoQuery. It is hard to trust the validity of the conclusions given the selected datasets.

* For the experimental conditions where most gains are seen (synthetic data), it seems that the entity types used convey extra knowledge about the synthetic data generation process, which is not available otherwise. Providing extra task information such as giving the generating grammar for SCAN to language models as an input has already been shown very useful before.

* Ablations: for the auxiliary loss method, it would be interesting to compare to using an auxiliary task that uses self-supervised training on the task unlabeled data. Such task adaptive pre-training has been found useful before and it is would be interesting to see whether predicting types is more useful than just predicting text from the task, especially for the tasks based on natural data.

* Computational expense: some of the methods increase the inference cost about 2x  – e.g. the method computing contextualized embeddings for two sequences. This is also the method that often performs well. A natural question would be whether moving to a larger pre-trained model would have a similar or larger effect.

---

> ### Author Response · Authors · 2022-07-20
> **Initial response**
>
> Thank you very much for taking the time to read our paper and writing constructive feedback.
> We especially appreciate the fact that you recognized that our paper is well written, has a lot of details on the methods and results, and that we have been very honest about our results.
>
> **About clarity**:
>
> We’ve updated Section 1 & 3 of the paper to address your clarification questions. In particular:
> - about the way we integrate the entity types: indeed, we do not pre-train on unlabeled text because the models we use are already pre-trained on large amounts of unlabeled text. Instead we use a fixed pre-trained entity tagger and then do end-task fine-tuning. see updated Sec1:
> > To construct the abstract representation incorporated into TLMs, we leverage entity type information given by fixed pre-trained models. This allows for automatic and reproducible data processing. In general, our approach is the following: given an input sequence, we use an entity tagger to label entity types in the sequence. We then use these labels to construct a copy of the original sequence in which all entities are replaced by their corresponding entity types. This new sequence can then be used as extra input or as extra training signal to the model.
> - Sec 3.3, abstraction as an auxiliary task: $X_s$ and $Y$ have indeed different lengths. We updated the notation of our equation. Regarding the suggestion about adaptive pre-training, we are not sure to understand what is meant by "*using an auxiliary task that uses self-supervised training on the task unlabeled data*”. Can you give an example or a reference to previous work that does this?
> - P2: by “efficient” we indeed meant “effective”. This was corrected.
> - Sec 3: we removed the formulation “such as” since we indeed used Spacy tagger.
>
> **About GeoQuery**:
>
> We will try to run experiments on this additional dataset in the coming days and include those results as soon as possible.
>
> **About the discussion on computational cost**:
>
> We briefly describe the additional number of parameters at the end of each subsection 3.1, 3.2, 3.3. Overall we did not find significantly longer training time for all approaches except for the dec-loss model that was training on average 2 times longer than the other methods. The most important factor in training time was the dataset used rather than the method used. Models trained on natural language datasets CoQA and HotpotQA took days to train while models trained on CLUTRR and ProofWriter took a few hours. We added this information to our limitations paragraph on the last page.
>
> We hope these clarifications will improve our paper and your understanding. We will add GeoQuery results in the coming days, until then don’t hesitate to continue the discussion on the above points if you have any questions.

---

> ### Author Response · Authors · 2022-08-10
> **Geoquery results**
>
> Hi,
>
> Thank you for your patience in getting these results.
>
> We've trained our models on two different splits of the Geoquery benchmark described in [Compositional Generalization and Natural Language Variation: Can a Semantic Parsing Approach Handle Both?](https://aclanthology.org/2021.acl-long.75) (Shaw et al., ACL 2021).
> In particular, they generated two splits focusing on compositional generalization: a split based on query length and a Target Maximum Compound Divergence (TMCD) split, each consisting of 440 train and 440 test examples.
>
> We trained our models on both of these and evaluated the exact match of the predicted output of our models.
> Below are our average results based on 3 random seeds.
>
> |  | LENGTH_split | TMCD_split |
> |---|---|---|
> | **no-abstraction** | 24.24% (+- 0.001) | **36.06%** (+- 0.001) |
> | **emb-sum** | **27.05%** (+- 0.000) | 34.47% (+- 0.001) |
> | **emb-cat**  | 15.91% (+- 0.002) | 18.56% (+- 0.001) |
> | **enc-sum** | 18.64% (+- 0.002) | 35.08% (+- 0.001) |
> | **enc-cat**   | 17.50% (+- 0.000) | 27.58% (+- 0.001) |
> | **dec-loss** | 24.09% (+- 0.000) | 30.38% (+- 0.001) |
>
> We can see that for the length split, one abstraction model performs better than the baseline (no-abstraction), but on the TMCD split, models with abstraction are not better than the baseline.
> However, we believe this dataset may not be a good candidate to showcase if abstraction is useful for 2 reasons:
>
> First, all examples in the dataset have a few number of entities (2 max):
>
> | n.of entities | n.of examples | percentage of the dataset |
> |---|---|---|
> | 0 | 207 | 23.5% |
> | 1 | 667 | 75.8% |
> | 2 | 6 | 0.7% |
> | **Total** | **880** | **100%** |
>
> This makes each abstract input very similar to the original input. Thus the abstract sequence provides very little information to the model.
>
> Second, the dataset as a whole contains a few number of entities. 12% of tokens per input are tagged as entities. This is a relatively small number compared to the other datasets as we can see below:
>
> Fraction of tokens per input in each dataset identified as entities by spacy NER:
> | Data | avg (+-std) |
> |---|---|
> | CLUTRR | 0.23 (+- 0.043) |
> | Proofwriter | 0.32 (+- 0.002) |
> | HotpotQA | 0.37 (+- 0.085) |
> | CoQA | 0.18 (+- 0.095) |
> | Geoquery | 0.12 (+- 0.080) |
>
> We will include the table above in our revised version.
> As for the Geoquery result table, we will include these results in the appendix of our revised version for further experimental analysis.
>
> Thank you.

---

### Review · Reviewer_Bo31 · 2022-07-26

**Summary Of Contributions:**

# Summary of Contributions
The main contribution of this paper is investigating whether --- and to what extent --- injecting entity type information is useful for improving the reasoning capability of generative Transformers. To that end, this paper proposes three intuitive approaches for injecting entity type information: (i) as additional input embeddings; (ii) as a separate, extra sequence to encode (in addition to the standard word sequences); and (iii) as an auxiliary prediction task at training time, which --- unlike the other two approaches --- only requires the entity type information at training time, and not at test time.

Experiments to assess the benefits of entity type information are done on two domains with formally defined logical reasoning settings (CLUTRR & ProofWriter), and on two reasoning-related natural language datasets (HotpotQA & CoQA). The main findings suggest that entity type information is much more beneficial for the first kind of tasks (i.e. for tasks with formally defined logical reasoning settings), whereas the same techniques only achieve minor gains for the reasoning-related natural language datasets. All in all, this finding contributes to a better understanding of where exactly a more explicit modelling of entity type information is beneficial for different types of reasoning tasks.

# Overall Assessment
Overall, the paper addresses an interesting and important research question: Given the difficulty of reasoning-based tasks for generative Transformer models, what sort of additional information can help improve them? This paper proposes a reasonable hypothesis that including entity type information can help improve reasoning ability, and then proceeds to validate this research hypothesis through 3 different methods on 4 diverse tasks. The paper is also generally clear and well-written.

Despite the strengths, I have two major concerns regarding the paper in its current form. These issues are elaborated more in the strengths and weaknesses section below, but can be briefly described as follows.
1. First, there is a strong possibility that the more minor improvements within the natural language tasks are due to coreference resolution and/or entity type tagging errors, which can introduce and propagate noise into the Transformer. Even the current best coreference resolution and/or entity type taggers are imperfect within their domain (and also potentially bounded by a limited sequence length), and their performance could be substantially worse for out-of-domain datasets. The paper makes an attempt to address this in the "Discussion" section on page 10, but it falls short because: (i) it only examines annotation errors on the CLUTRR dataset, which is not exactly natural language (where entity tagging is much more difficult and ambiguous), and on which the model performs well already; (ii) it does not discuss or explore any techniques for mitigating these errors (see point 1 of the weakness section below for some suggestions).
2. The paper also does not try other potentially better methods that have been successfully used for injecting syntactic structure information into Transformer models, and enable cross-attention between the word sequence information and the entity type sequence information (see point 2 of the weakness section below).
3. This is more minor, but having statistical significance tests and/or reporting mean and standard deviation of the performance on each task would improve the credibility of the findings, particularly on the natural language tasks where the gains are fairly small.

**Broader Impact Concerns:**

No broader impact concern from my end.

**Requested Changes:**

1. **Critical**: A more detailed analysis about potential tagging errors, especially in the context of the natural language tasks (CoQA, HotpotQA) where entity tagging is highly ambiguous, prone to errors, and potentially requires long-range context information. This can be done by: (i) manually inspecting a small subset of the entity tagger's outputs, and getting an estimate of the tagger's performance; and/or (ii) using established techniques such as tri-training in order to get high-confidence tagging predictions.

2. **Strongly recommended**: Try the approach of Choe and Charniak (2016); Sartran et al., (2022) that jointly aims to predict the linearised sequence of words and tags (see point 2 of the weakness above). This would enable cross-attention between the word information and the entity information, unlike the current approaches.

3. **Critical**: Add comparisons with the performance of previously published models (ideally that have comparable model sizes and training setups with the proposed models). Furthermore, add statistical significance and/or report mean & standard deviations to all reported numbers. Both of these (external comparison and statistical significance testing) can improve the credibility of the findings.

4. **Strongly recommended**: add the following links to the relevant prior work: (i) emergent reasoning ability of very large scale language models, e.g. Palm (Chowdhery et al., 2022); (ii) prior work on multi-task auxiliary loss for injecting syntax into LMs & other downstream tasks, e.g. (Swayamdipta et al., 2018; Eriguchi et al., 2017; Nadejde et al., 2017, and others); (iii) adding syntactic information to the input embedding of Transformers (Sundaraman et al., 2019).

5. Clarification questions (**strongly recommended** to clarify in subsequent versions of the paper):
- Comparing Tables 2 and 3, it seems that the best-performing variant in each table is different from one another. Do you have insights or conjectures on why this is the case?
- Would an unsupervised learning approach that aims to **jointly** induce the entity type information and the standard generative modelling loss be a promising path for future work? This would eliminate the need to have a "pipeline" system that predicts the entity type information first, and thus mitigate the risk of the wrong entity type information being propagated to the Transformer model.
- I find the following sentence at the end of page 3 difficult to read and interpret, please consider rephrasing it and/or provide illustrative examples: "Although this may look similar to ... such ID across examples and across epochs."
- In Section 3.2, it seems that the encoder is shared across the 2 different sequences (word sequences & entity type sequences). What happens if you use two separate encoders instead? This means less information sharing, but it might be compensated by an implicit ensembling effect and an increase in the model capacity (by virtue of having more parameters).

6. Typos:
- In page 3, "in order to encode **syntaxt** trees"
- In page 8, "etc..." -> should there be only 1 period here?

**Strengths And Weaknesses:**

# Strengths
1. The paper addresses an interesting and important research question: Given the difficulty of reasoning-based tasks for generative Transformer models, what sort of additional information can help improve them? More broadly, answering this question enables the community to better understand whether, and to what extent, features that are extracted by "classical" NLP pipelines (e.g. syntax, coreference and entity information, semantic information, etc.) are still relevant in the current era of large, end-to-end Transformer models, and if so, how we should best incorporate these types of information.

2. The paper observes substantial improvements for tasks that involve formally defined reasoning settings, which can have downstream implications (e.g. writing new proofs and theorems, or validating existing ones, and maybe also for programming language & semantic parsing domains, etc.).

3. The paper is generally clear, well-written, and easy-to-follow.

4. The proposed approaches are simple and intuitive, which could lower the barrier for more widespread adoption.

# Weaknesses

1. There is a strong possibility that the more minor improvements within the NLP tasks are due to coreference resolution and/or entity type tagging errors, which can introduce and propagate noise into the Transformer. Even the current best coreference resolution and/or entity type taggers are imperfect within their domain (and also potentially bounded by a limited sequence length, e.g. 512 for BERT-based ones, and thus unable to condition on the entire relevant past information such as the whole document). It is also reasonable that their performance would be substantially worse when annotating out-of-domain datasets, which is the case here in this work. The paper makes an attempt to address this in the "Discussion" section on page 10, but in my opinion it falls short because: (i) it only examines annotation errors on the CLUTRR dataset, which is not exactly natural language (where entity tagging is much more difficult and ambiguous), and on which the model performs well already (unlike, CoQA or HotpotQA); (ii) it does not discuss or explore any techniques for mitigating these automatic annotation errors, such as tri-training, which has been widely used in the syntax & structure literature within NLP (Weiss et al., 2015; Choe & Charniak, 2016).

2. The paper also does not try another, potentially better method that has been successfully used for injecting syntactic structure information into Transformers, and enables cross-attention between the word sequence information and the entity type sequence information (unlike the paper's second approach, that **separately** encodes the entity type sequence and the word sequence). This technique is done by simply combining the entity type information and the word in a single sequence, in the same fashion as the linearised syntactic tree model of Choe and Charniak (2016), which has been more recently explored in generative Transformers by Sartran et al. (2022), leading to improvements over the standard, word-level Transformers. A straightforward application to the entity type information case would be "Alex PERSON_23 Andra PERSON_23 Smith PERSON_23 is [non-entity] the [non-entity] wife [non-entity] of [non-entity] Bob PERSON_11". This would naturally result in a longer sequence, but given (i) its effectiveness in another related domain (syntax), and (ii) the ability of the model to do cross-attention between word and entity-level information, and (iii) overall simplicity & straightforward extension over the paper's methods, it is worth trying.

3. It would be great to add statistical significance tests and/or reporting mean and standard deviation performance to account for noise in the reported performance, as the improvements for the NLP tasks are rather small.

4. There is a lack of comparison with numbers from **external** work (e.g. external/previously published models that have comparable model size & training/test setups as this work), which would improve the credibility of the findings and the author's implementations.

5. Although the approaches are simple and easy-to-implement, they are not particularly novel, and in fact have been used in other domains, e.g. the auxiliary task has been done in the case of syntactic scaffolds (Swayamdipta et al., 2018) and joint parsing and machine translation (Eriguchi et al., 2017); the input embedding approach has been explored --- also in the context of syntax --- by Sundaraman et al. (2019).

More minor presentational suggestions are provided in the "Requested Changes" section below.

---

> ### Author Response · Authors · 2022-07-28
> **Initial Response**
>
> Thank you very much for your very detailed review and all the feedback you provided. We especially appreciate the fact that you clearly understood our motivation and research question and that answering this question enables the community to better understand whether features that are extracted by "classical" NLP pipelines are still relevant in the current era of large, end-to-end Transformer models, and if so, how we should best incorporate these types of information.
>
> **critical point #1: tagger accuracy**
> Thank you for mentioning tri-training. However, in the interest of time, we will prioritize a manual inspection of the performance of the entity tagger on the natural datasets (HotpotQA & CoQA) as you also suggested. This will show to what degree your initial hypothesis (that the tagger performance is at play) holds.
>
> **strongly recommended point #1: extra experiments**
> Thanks for suggesting yet another way to incorporate entity tags information into our models. We will try to run experiments with this new strategy if time allows, although we will first prioritize adding more analysis for the current experiments.
>
> **critical point #2: statistical significance**
> It is unfortunately hard to compare against previous work as we use the datasets in ways that prior work did not. However, we will run multiple predictions in order to provide a mean and standard deviation for our experiments.
>
> **Short answers**:
>
> - Thank you for suggesting many relevant prior work. We will make sure to update our paper accordingly.
>
> - For CLUTRR results (Table 2) we test longer reasoning lengths (up to 10 steps), making the input sequence much longer than what is seen during training. This is because in CLUTRR, to make long reasoning chains the input family story gets longer. This penalizes the dec-loss model since it has to generate the long input (abstracted, but still longer than seen during training). On the other hand, in ProofWritter (Table3) the reasoning depth is not correlated with the length of the input like in CLUTRR, thus the dec-loss model is not as penalized, resulting in great performance.
>
> - We believe that an unsupervised learning approach that aims to jointly induce the entity type information and the standard generative modeling loss is an interesting suggestion for future work. It could be a promising path, however it may be tricky to properly make the entire system (both language modeling and NER) work. We also suspect that the advantage of this approach will depend on whether or not the model needs to generalize to longer reasoning steps with similar entities as seen during training. We found recent work that performs unsupervised NER and provides some related work: CycleNER: An Unsupervised Training Approach for Named Entity Recognition (https://dl.acm.org/doi/10.1145/3485447.3512012)
>
> - We will rephrase as best as we can the third paragraph of Section 3 (bottom of page 3) as we realize it can be confusing. Overall, we mean that after seeing many examples, the fact that we use n>1 tokens per entity type will have the same abstraction effect as using n=1 token per entity type.
>
> - Although an interesting question, we didn't try to use two separate encoders due to the number of parameters being greatly increased. It would have resulted in much longer training time and required greater compute resources. We also agree that it would increase capacity.
>
> - Thanks for flagging the typos, they will be corrected.
>
> We hope the above action plan will improve our paper, until then don’t hesitate to continue the discussion if you have any questions. Thanks again for your detailed review.

---

> > ### Comment · Reviewer_Bo31 · 2022-08-17
> > **Reply to Initial Response**
> >
> > Thank you for the clarifications and for the revised version of the paper, which improves over the original one by taking into account various reviewers' feedback (e.g. the GeoQuery results, reporting mean and standard deviations, adding a manual inspection of the tagger performance, etc.).
> >
> > I do have 2 more clarification questions that I hope the authors can clarify.
> >
> > 1. Regarding the manual inspection of the tagger performance, it was stated in one of the comments that "we performed a manual inspection of the performance of the entity tagger for all our datasets (see 'correct entities' column from table below). Note that we only measure the proportion of tagged entities that are correct. We did not count entities that were ignored by the tagger.". Does this mean that the tagger performance is reported in terms of **precision**, rather than **recall**? Is there a way to quantify the recall of the system as well?
> >
> > 2. For the CLUTRR results, it was shown that some methods were more robust to 10% than others. On the flip side, does this mean that those approaches make **less use** of the entity information (since having the wrong entity information does not change the model's predictions much)? One way to test this would be to increase the amount of noise; for instance, if the approaches still perform similarly even with a very high amount of noise, then this indicates that having the right entity information is not important (which would be counter-intuitive, and indicates that the model does not make much use of the provided entity information).

---

> > > ### Author Response · Authors · 2022-08-18
> > > **Reply to 2 more questions**
> > >
> > > Thank you for recognizing the improvement over the original version of the paper.
> > > We answer your two questions below:
> > >
> > > 1. Thank you for pointing it out, you are entirely correct, we previously reported the precision of the tagger. In order to measure the recall, we ran another manual inspection of the same examples and counted unrecognized entities. From this, we estimated the following Recall numbers: CLUTRR: 100%, ProofWriter: 100%, **HotpotQA: 93.2% (+- 0.069), CoQA: 94.8% (+- 0.086)**. We will update Table6 in the manuscript to reflect this additional evaluation.
> > >
> > > 2. This is an interesting hypothesis. We believe that the model is still using this information since the performance of the "with-abstraction" models are relatively different than the "no-abstraction" model but we can run additional experiments with 25%, 50%, 75% noise to be sure. This will take some time so we might defer those experiments as minor revisions after the decision of the Action Editor.
> > >
> > > Thank you again for the great feedback you provided in this work.

---

### Review · Reviewer_THgz · 2022-07-27

**Summary Of Contributions:**

In this paper,  the authors investigate the research question that whether or not injecting the entity-type information can help reasoning tasks in NLP with generative Transformer models. They use three straightforward methods to inject the entity-type information: 1) use them as additional input embeddings, 2) encode them with additional individual sequences separately from the token sequences, and 3) create auxiliary tasks predicting the entity types for joint training. The authors found that entity-type info can only benefit the reasoning tasks that require/have formal logic constraints (e.g., CLUTRR and ProofWriter), while they are not very useful for tasks like HotpotQA and CoQA which do not have formal logic elements.

**Broader Impact Concerns:**

No.

**Requested Changes:**

- Section 4 can include more case studies and qualitative analysis. (See more in the Weakness part.)
- Focus on tracking the model behavior changes before and after the injection of entity-type info. This can largely improve the analysis section and provide more insights to the readers.
- Use more visualization to discuss the results so that the analysis and conclusions are more accessible to readers.


**Strengths And Weaknesses:**



Strengths

- The research question is interesting to the community of entity-centric reasoning and logical reasoning. Injecting entity-type info into neural language models has been tried in many places and there is no such detailed analysis.

- The authors create a universal framework for comparing different injecting methods and different tasks for Transformer models. I believe the codebase for this evaluation framework can be used for other people to continue exploring other methods and evaluating more extensively. This is also the first time we can get a comprehensive understanding of the potential benefits of entity-type info for reasoning.

- The paper is well-written and introduces the background knowledge, motivation, methods, and evaluation with a nice presentation for most of the parts.

Weakness
- I would like to see more analysis with real case studies on how the injection of entity-type info really changes the model behaviors, and thus enhance the reasoning performance. However, the main experiment section is mostly empirical results. Also, I think it is more beneficial if the authors can include some categorical analysis on how these behavior changes are related to the injected entity types, such that the paper can provide more insights for future research.

- The overlap across the injection methods is not analyzed. Given the above suggestion, once there are more detailed, categorical case studies on the model behavior changes, this should be easier to analyze. Then, we can clearly see the similarities and differences between these injection methods.

- The error analysis is also important. Before and after the injection of entity-type info, do the models correct their errors in an expected way? Or do they correctly predict answers to some new examples but make mistakes on even simpler questions? Without detailed error analysis and tracking of the model behavior, the empirical results themselves are not convincing to me at all.

---

> ### Author Response · Authors · 2022-07-28
> **Initial response**
>
> Thank you for taking the time to read our work and write your feedback. We especially appreciate the fact that you recognize that our work is the first to get a comprehensive understanding of the potential benefits of entity-type info for reasoning.
>
> We will provide output examples of our different approaches and some comparative analysis in the next few days.
> We are not sure to understand what you mean by “categorical analysis”. If you mean compare the output of the different models based on the injection strategy, we will do that in our comparative analysis. Similarly for the "error analysis" we will touch on that when comparing the different model outputs.
> We hope this will improve our paper, until then don’t hesitate to continue the discussion if you have any questions.

---

> ### Author Response · Authors · 2022-08-20
> **output examples**
>
> Hi,
>
> We present below 5 output examples of our models on the CLUTRR level 3 test set.
>
> | input | no abstraction | emb-sum | emb-cat | enc-sum | enc-cat | dec-loss |
> |-----------|-----------|-----------|-----------|-----------|-----------|-------------|
> | Jonathan is the father of Anne . Anne is a aunt to Stephanie . Stephanie is the daughter of Veronica . How is Veronica related to Jonathan ? | Jonathan is Veronica 's father [OK] | Jonathan is Veronica 's father [OK] | Jonathan is Veronica 's uncle [X] | Jonathan is Veronica 's father [OK] | Jonathan is Veronica 's father [OK] | Jonathan is Veronica 's father [OK] |
> | Stephanie is the granddaughter of Jonathan . Bryant is Stephanie 's father . Eric is a son of Bryant . How are Jonathan and Eric related to each other ? | Eric is the grandson of Jonathan [OK] | Eric is Jonathan 's grandson [OK] | Eric is Jonathan 's grandson [OK] | Eric is Jonathan 's grandson [OK] | Eric is Jonathan 's grandson [OK] | Eric is Jonathan 's grandson [OK] |
> | Joyce has a daughter called Stephanie . Betty is a grandmother to Stephanie . Paul has a wife who is Joyce . For Paul , who is Betty ? | Betty is Paul 's mother-in-law [OK] | Betty is Paul 's mother-in-law [OK] | Betty is Paul 's aunt [X] | Betty is Paul 's mother-in-law [OK] | Betty is Paul 's mother-in-law [OK] | Betty is Paul 's mother-in-law [OK] |
> | Bryant is a son of Jonathan . Jonathan is the husband of Betty . Craig is a son of Bryant . Who is Craig from the point of relation of Betty ? | Craig is Betty 's son [X] | Betty has a grandson who is Craig [OK] | Craig is Betty 's grandson [OK] | Betty has a grandson who is Craig [OK] | Craig is Betty 's son [X] | Craig is the son-in-law of Betty [X] |
> | Stephanie is a granddaughter to Betty . Anne is Bryant 's sister . Bryant is a brother of Stephanie . Who is Anne from the point of relation of Betty ? | Anne is Betty 's granddaughter [OK] | Anne is Betty 's granddaughter [OK] | Anne is Betty 's granddaughter [OK] | Anne is Betty 's granddaughter [OK] | Anne is Betty 's granddaughter [OK] | Anne is Betty 's granddaughter [OK] |
> | **score** | **4/5** | **5/5** | **3/5** | **5/5** | **4/5** | **4/5** |
> | | | | | | | |
> | score on lvl.3 test set reported in Table2 of paper | 84.4% | 85.3% | 60.0% | 94.8 % | 86.1% | 74.7% |
>
> Out of these 5 examples, the emb-cat model makes 2 mistakes (line1 & line3), the no-abstraction, enc-cat, dec-loss models make 1 mistake (line4) and the emb-sum and enc-sum models make 0 mistake.
>
> The emb-cat model mistakes (line1 & line3) are due to the prediction of uncle/aunt relations instead of father/mother-in-law respectively. It is interesting to note that these relations are similar in the sense that they all link one generation to the one just above.
>
> It is also interesting to note that the example in which the no-abstraction, enc-cat and dec-loss models fail (line4), the emb-cat model on the other hand answers correctly. In this specific example the correct relationship is "grandson" which links one generation to 2 generations below, however the mistakes made by the failing models only link one generation to 1 below (son, son, son-in-law).
>
> We will include this small error analysis in the appendix of the final version of the paper.

---

### Author Response · Authors · 2022-08-11
**Important changes to the revised version**

Thank you to all reviewers for taking the time to provide useful feedback.

We would like to point to your attention the ongoing work towards this submission, as we incorporated some of your required changes.

Currently in the revised PDF:
- we clarified a bit the strategy we use in Section 1
- we reformulated our loss in Section 3.3
- we rephrased the third paragraph in Section 3 for better clarity
- we explain why the best model in Table2 & Table3 are not the same in Sections 4.1 & 4.2
- we added additional relevant prior work in Section 2
- we ran prediction with multiple random seed to include mean & standard deviation in Table 3 & Table 4

Done work, but not in the PDF yet:
- we ran experiments on another dataset (GeoQuery). Results will be appended in an appendix.
- we computed the average proportion of tokens being tagged as entities per input sequence for all our datasets (see 'entity tokens' column from table below).
- we performed a manual inspection of the performance of the entity tagger for all our datasets (see 'correct entities' column from table below). Note that we only measure the proportion of tagged entities that are correct. We did not count entities that were ignored by the tagger.

| Data | entity tokens | correct entities |
|---|---|---|
| CLUTRR | 22.2% (+- 0.019) | 100% |
| Proofwriter | 36.0% (+- 0.036) | 100% |
| HotpotQA | 37.0% (+- 0.085) | 84.3% (+- 0.137) |
| CoQA | 17.7% (+- 0.095) | 82.7% (+- 0.083) |

We will include the table above in a new section in the paper in which we will discuss in more details our results and present working & failing cases.

---

> ### Author Response · Authors · 2022-08-15
> **New Revision**
>
> As mentioned previously, we now added a new section (between experimental results and conclusion) in which we discuss our results and present further analysis. In particular we added:
>
> - the average proportion of tokens being tagged as entities per input sequence for all our datasets.
> - a manual inspection of the performance of the entity tagger for all our datasets.
> - CLUTRR results with weaker entity tagger.
>
> See updated PDF for all the details.

---

### Author Response · Authors · 2022-08-20
**Recap**

Thank you again for all the feedback we received. We would like to recap the reviewing and revision activity of this work.
We’ve compiled below a list of action items from all reviews. Some of them are already included in the latest version of the manuscript and we are willing to do the remaining items for the camera-ready version of the paper after the decision from the action editor.

legend:
- [X]  Done. in PDF.
- [_] will be provided for the camera-ready version

| **Reviewer** | **Action Item**    | **Status**  |
|-------------------|------------------------|---------------|
| THgz         | error analysis on output examples          | [x] |
| Bo31         | CLUTRR experiments with 25% 50% 75% noise | [_] |
|              | inspection of the performance of the entity tagger  | [x] |
|              | experiments with new strategy    | [_] |
|              | report mean and standard deviation    | [x] |
|              | include some relevant prior work | [x] |
|              | explain difference between best model in Table2 & 3 | [x] |
|              | rephrase Section 3               | [x] |
| 7k7T         | extra experiments on GeoQuery    | [x] |
| M3sQ         | report additional statistics     | [x] |
|                | additional analysis section | [x] |

---

### Decision · Action_Editors · 2022-09-19

**Recommendation:** Accept with minor revision

**Comment:**

This work presents an analysis of different ways of incorporating entity information in transformer models either through architectural changes or loss modifications. Apart from the different ways of incorporating entity information, the authors also consider a range of evaluations benchmarks covering:
* logical reasoning (CLUTRR and ProofWriter) that stress-test compositional reasoning through synthetically-generated examples
* multi-hop reasoning (HotpotQA) and
* conversational reasoning (CoQA).

The main findings are that entity abstraction helps for compositional reasoning but provides little to no benefit on realistic use cases that require perhaps more world knowledge. While all reviewers, myself as well, agree this is an important question, I believe there are a couple of things missing currently from the manuscript that would help complete the picture – I recommend this as a minor revision case.

The minor revisions should cover the following things:
1) As the method relies on the availability and performance of an entity-tagger, it is important to simulate performance and comparison to baseline with varying degrees of noise.
2) While it is true that authors do not wish to obtain SoTA, it is difficult to assess the extent of improvements wrt previous work. It is important for each dataset to make an effort to contextualize these results as much as possible.
3) An important question is why the entity abstraction does not help in the more realistic datasets. The authors provide a couple of hypotheses, but it is important to perhaps make an effort to provide a more in-depth answer, in the context of this being an analytical study. For example, what happens if one looks at the predictions that are the same/differ with respect to the baseline, can some pattern be observed? Or, how does the accuracy of the tagger look like in a small subset of data, can this explain the difference? Even if a definate answer is not easy to be reached, perhaps some hypotheses can still be tested and eliminated.

---

> ### Author Response · Authors · 2022-10-27
> **Camera Ready version**
>
> Thank you for your feedback and positive recommendation.
> We took the time to address all of your suggestions below:
>
> 1. Regarding the impact of the performance of the entity tagger: we simulated weaker tagging performance with varying degrees of noises (25%, 50%, 75%) and compared results to the baseline model. Results on CLUTRR and ProofWriter are reported in Figure 1 of the camera ready version of the paper.
>
> 2. Regarding contextualizing with respect to previous work: we contextualized our HotpotQA and CoQA results by referring to previous model performances in the last paragraphs of section 4.3 and 4.4 respectively.
>
> 3.  **Q1**: what happens when one looks at predictions that are the same and that differ with respect to the baseline model? As suggested, we have provided an analysis of predictions that are the same and that differ from the baseline for CLUTRR and HotpotQA in Appendix F1 and F2 respectively.
> **Q2**: what does the accuracy of the tagger look like on a small subset of the data? As recommended, we have added the accuracy of the tagger on a small subset of all datasets in Table 6 of the camera ready version of the paper.
> **Regarding hypotheses related to understanding our results**: we observed on CLUTRR and ProofWriter that with higher noise levels in the entity tagger, the overall model accuracy is reduced to the baseline model or even below. Thus the weaker tagger accuracy on HotpotQA and CoQA is part of the explanation.
> We also make a number of interesting observations about HotpotQA in Appendix F2.
> Overall, we believe three factors are influencing our results on the more natural tasks: (i) the effect of a weaker entity tagger performance as shown in Table 6 and Figure 1, (ii) the depth of reasoning being relatively shallow and the same across training and testing sets as observed in Appendix D and F.2, (iii) the natural and conversational language being further away from a formal language.
>
> We thank again all the reviewers for their time and suggestions.